# circCDK13-loaded small extracellular vesicles accelerate healing in preclinical diabetic wound models

Qilin Huang[1,2,10], Ziqiang Chu[2,3,10], Zihao Wang[2,3,4,10], Qiankun Li[5], Sheng Meng[2,3], Yao Lu[5], Kui Ma[2,3], Shengnan Cui[2,6], Wenzhi Hu[2], Wenhua Zhang[2], Qian Wei[2], Yanlin Qu[2], Haihong Li[7] ✉, Xiaobing Fu ⓘ[2,3,8,9] ✉ & Cuiping Zhang ⓘ[2,3,8] ✉

Chronic wounds are a major complication in patients with diabetes. Here, we identify a therapeutic circRNA and load it into small extracellular vesicles (sEVs) to treat diabetic wounds in preclinical models. We show that circCDK13 can stimulate the proliferation and migration of human dermal fibroblasts and human epidermal keratinocytes by interacting with insulin-like growth factor 2 mRNA binding protein 3 in an N6-Methyladenosine-dependent manner to enhance CD44 and c-MYC expression. We engineered sEVs that overexpress circCDK13 and show that local subcutaneous injection into male db/db diabetic mouse wounds and wounds of streptozotocin-induced type I male diabetic rats could accelerate wound healing and skin appendage regeneration. Our study demonstrates that the delivery of circCDK13 in sEVs may present an option for diabetic wound treatment.

Nowadays, the living standard of citizens is improving due to the progress of society, but a series of problems are emerging at the same time. For example, the number of diabetic patients is increasing steadily year by year. Chronic non-healing wounds are one of the major complications of diabetes, seriously affecting the patient's quality of life and imposing a heavy economic burden on families and society[1]. Suboptimal glycemic control is tightly linked to the occurrence and development of diabetic skin lesions. Advanced glycation end products (AGEs) are formed by non-enzymatic reactions between reducing sugars and a variety of macromolecules such as proteins, lipids, and nucleic acids[2]. With the stimulation of long-term hyperglycemia, the AGEs level is significantly higher in diabetic patients than that in normal individuals. Accumulating evidence suggests that the presence and accumulation of AGEs in diabetic wounds affect the functional status of skin repair cells, such as epidermal keratinocytes and dermal fibroblasts, resulting in delayed wound healing[3,4]. Despite recent advances in the comprehension of impaired wound healing in diabetes, little is known about the molecular mechanisms responsible for the decreased function of skin repair cells, and the current therapeutic efficacy of existing interventions is not favorable.

Circular RNAs (circRNAs) are novel members of the highly conserved family of endogenous non-coding RNAs. Structurally, circRNAs form a covalent closed-loop structure without 5′ caps and 3′ tails[5], which makes them hold a longer half-life and more resistance to RNase

[1]Tianjin Medical University, No. 22, Qixiangtai Road, Heping District, Tianjin 300070, China. [2]Research Center for Tissue Repair and Regeneration affiliated to the Medical Innovation Research Department, PLA General Hospital, 51 Fucheng Road, Haidian District, Beijing 100048, China. [3]Research Unit of Trauma Care, Tissue Repair and Regeneration, Chinese Academy of Medical Sciences, 2019RU051, 51 Fucheng Road, Haidian District, Beijing 100048, China. [4]Chinese PLA Medical School, 28 Fuxing Road, Haidian District, Beijing 100853, China. [5]Department of Tissue Repair and Regeneration, The First Medical Center, Chinese PLA General Hospital, Beijing 100853, China. [6]Department of Dermatology, China Academy of Chinese Medical Science, Xiyuan Hospital, Beijing 100091, China. [7]Department of Burns and Plastic Surgery, the Seventh Affiliated Hospital of Sun Yat-sen University, Shenzhen, Guangdong Province 518055, China. [8]PLA Key Laboratory of Tissue Repair and Regenerative Medicine and Beijing Key Research Laboratory of Skin Injury, Repair and Regeneration, 51 Fucheng Road, Haidian District, Beijing 100048, China. [9]Innovation Center for Wound Repair, West China Hospital, Sichuan University, Chengdu, Sichuan Province 610041, China. [10]These authors contributed equally: Qilin Huang, Ziqiang Chu, Zihao Wang. ✉e-mail: lihaihong1051@126.com; fuxiaobing@vip.sina.com; zcp666666@sohu.com

R than linear RNAs[6]. Extensive studies have revealed that circRNAs exert biological functions under a series of physiological and pathological conditions by acting as protein scaffolds, microRNA sponges, and protein templates[5]. Recent studies have shown that the progression and pathogenesis of diabetic wounds are critically influenced by a wide array of endogenous circRNAs. Toma et al. discovered that hsa-CHST15_0003 and hsa-TNFRSF21_0001, which are upregulated in venous ulcers, hampered epidermal keratinocyte migration[7]. Another study showed that hsa_circ_0084443 was up-regulated in diabetic foot ulcer (DFU) and inhibited keratinocyte migration[8]. Theoretically, the up-regulated circRNAs could be used as therapeutic targets for wound healing, and the down-regulated circRNAs, which have never been reported in diabetic wounds, could be explored as the promising therapeutic molecules. However, the administration of circRNAs requires a safe and effective delivery system, which is beneficial for clinical translation.

Small extracellular vesicles (sEVs) are lipid bilayer membrane-enclosed nanoscale particles ranging in size from 30 nm to 150 nm. sEVs have been recognized as critical intercellular messengers by transferring RNAs (such as microRNAs, lncRNAs, and circRNAs), DNAs, and proteins[9–11]. Compared to cell therapy, sEVs have many advantages as a cell-free therapy strategy, including higher stability, lower immunogenicity, fewer ethical issues, and lower risk of embolization and tumorigenesis[12]. Additionally, sEVs have great promise as efficient drug delivery vehicles for their properties, including excellent biocompatibility and intrinsic long-term circulatory capability, which are suitable for delivering therapeutic agents[13,14]. More recently, encapsulating circRNAs in engineered sEVs for disease treatment has been reported. Yang et al. identified that sEVs-mediated delivery of circSCMH1 could promote functional recovery in rodent and nonhuman primate ischemic stroke models[15]. Yu et al. demonstrated that engineered sEVs overexpressing circDYM alleviated depressive-like behaviors induced by chronic unpredictable stress[16]. Recently, accumulating studies have confirmed that mesenchymal stem cells-derived sEVs (MSC-sEVs) could promote wound healing[17]. Therefore, we sought to use MSCs to prepare engineered sEVs carrying circRNA, which might further enhance the efficacy of MSC-sEVs in promoting wound healing.

In the present study, we identify a CDK13-derived circRNA, hsa_circ_0079929, named circCDK13, which is downregulated in diabetic wounds and closely associated with the slow healing of diabetic wounds. Both gain-of-function and loss-of-function experiments confirm that circCDK13 plays a crucial role in regulating the proliferation and migration of human dermal fibroblasts (HDFs) and human epidermal keratinocytes (HEKs) by interacting with IGF2BP3 in m[6]A-dependent manner. Subsequently, we successfully construct engineered MSC-sEVs overexpressing circCDK13, namely circCDK13[OE]-sEVs. The results demonstrate that circCDK13[OE]-sEVs restore the functions of AGEs-induced HDFs and HEKs in vitro and accelerate diabetic wound healing in type II diabetic mice (db/db mice) and type I diabetic rats, and its ability to promote the proliferation and migration of HDFs and HEKs, as well as wound healing, is significantly stronger than that of natural MSC-derived sEVs (N-sEVs). Our data suggest that circCDK13[OE]-sEVs represent a promising therapeutic strategy for diabetic wound healing.

## Results

### circCDK13 was chosen as a promising therapeutic agent for diabetic wounds

circRNAs have been discovered as powerful actors of gene expression with critical functions in many diseases. To elucidate the role of circRNAs in the healing process of diabetic wounds, we analyzed a set of microarray data (GSE114248) published by Wang[8], and found that 111 circRNAs including circCDK13 were down-regulated in the DFU compared with normal wounds (NWs) (Supplementary Fig. 1a). Heat map of the top 40 circRNAs was shown in Fig. 1a and normalized expression

level of circCDK13 were calculated with the microarray data (Fig. 1b). Homology analysis was performed on the top 40 circRNAs and we found that 6 circRNAs including circCDK13 shared homology between Homo sapiens and mice. The homologous sequence of circCDK13 was shown in Supplementary Fig. 1b. Additionally, the lower expression of circCDK13 was also observed in the wounds of diabetic mice (Fig. 1c). To simulate the microenvironment of diabetic wounds in vitro, AGE-BSA, an alternative to AGEs commonly used in research, was produced by co-incubating D-glucose with bovine serum albumin (BSA) at 37 °C for 2 months (Supplementary Fig. 2a−c). After treated with AGE-BSA at different concentrations, HDFs and HEKs exhibited the decreased proliferation assessed by CCK8 assay (Supplementary Fig. 3a, b) and migration assessed by scratching test (HDFs: Supplementary Fig. 4a, c; HEKs: Supplementary Fig. 5a, c) and transwell assay (HDFs: Supplementary Fig. 4b, d; HEKs: Supplementary Fig. 5b, d). Among above 6 circRNAs, we found that three circRNAs, including circCDK13, were significantly downregulated in AGE-BSA treated HDFs and HEKs. Figure 1d, e demonstrated the downregulation of circCDK13 in AGE-BSA-treated HDFs and HEKs. Among the three downregulated circRNAs, circCDK13 was reported to play a significant role in the regulation of cellular proliferation and migration[18]. Therefore, we speculated that the downregulation of circCDK13 might be involved in the mechanisms of impaired wound healing in diabetes, implying circCDK13 was a promising therapeutic agent for diabetic wounds. The screening process of circCDK13 was summarized in Supplementary Fig. 1c.

### Characterization of circCDK13 in HDFs and HEKs

To clarify the origin of circCDK13, we searched circBank online database and identified that circCDK13 was a circular transcript generated from the back-splicing of the exon 2 of the CDK13 gene located on human chromosome 7 with a length of 660 nt, and the back splice junction site of circCDK13 was confirmed via Sanger sequencing (Fig. 1f). To eliminate the potential influence of trans-splicing, genomic rearrangements, or PCR artifacts on the observed head-to-tail splicing, convergent and divergent primers were designed to amplify CDK13 mRNA and circCDK13, respectively. Agarose gel electrophoresis of the PCR products revealed that circCDK13 was amplified by divergent primers in cDNA, but not in gDNA (Fig. 1g). Actinomycin D assay (Fig. 1h, i) and RNase R exonuclease digestion assay (Fig. 1j, k) further indicated that circCDK13 was more stable than CDK13 mRNA. Furthermore, fluorescence in situ hybridization (FISH) (Fig. 1l) and nuclear and cytoplasmic fractionation analysis (Fig. 1m, n) demonstrated that circCDK13 was predominantly localized in the cytoplasm. Collectively, these results implicated that circCDK13 in HDFs and HEKs is a stable circRNA generated from CDK13 by back-splicing.

### circCDK13 promotes the proliferation and migration of HDFs and HEKs in vitro

To further explore the biological significance of circCDK13 in wound healing, we selected two types of skin repair cells (HDFs and HEKs), which are very important in wound healing. Two siRNAs (Fig. 2a) targeting the back-splicing sequence of circCDK13 were used to knockdown the expression of circCDK13 in HDFs (Fig. 2b) and HEKs (Fig. 2c). CCK-8 assay was performed to show that the knockdown of circCDK13 resulted in the decreased viability of HDFs (Fig. 2d) and HEKs (Fig. 2e). Next, an overexpression vector of circCDK13 was constructed (Supplementary Fig. 5a, b) and then transfected into HDFs and HEKs (Fig. 2f) to overexpress circCDK13 (Fig. 2g, h). The results of CCK-8 assay show that the enforced expression of circCDK13 dramatically enhanced the viability of HDFs (Fig. 2i) and HEKs (Fig. 2j). Furthermore, the percentage of EdU-positive cells was decreased by circCDK13 knockdown and increased by circCDK13 overexpression in HDFs (Fig. 2k, Supplementary Fig. 6c, d) and HEKs (Fig. 2l, Supplementary Fig. 6e, f). Subsequently, scratch and transwell assays were performed to investigate the effect of circCDK13 on cell migration. The migration

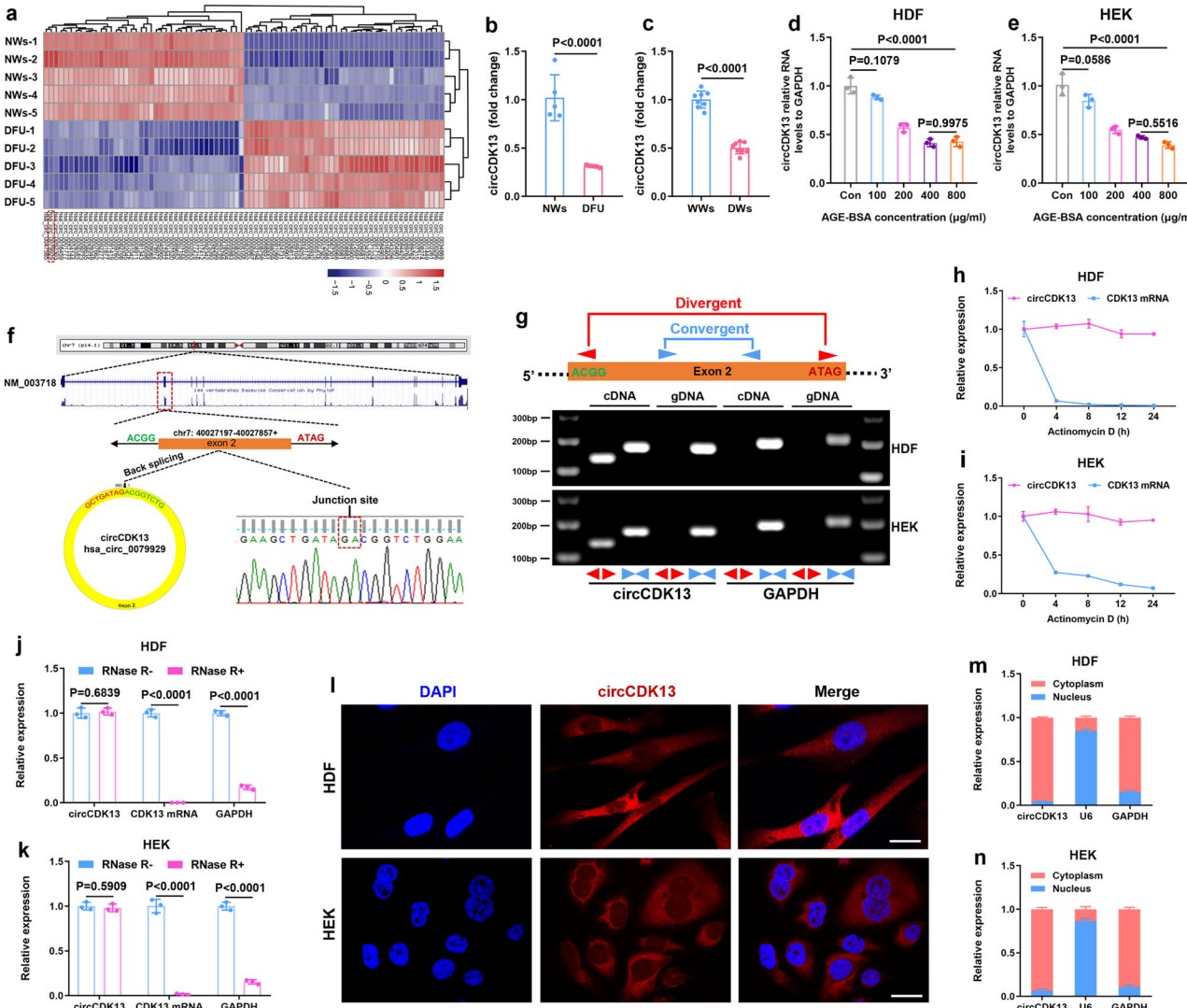

**Fig. 1 | circCDK13 was downregulated in diabetic wounds. a** Heat map of the top 40 circRNAs with significant differential expression. Red represents up-regulated circRNAs, blue represents down-regulated circRNAs in diabetic foot ulcer (DFU) vs normal wounds (NWs). **b** The normalized expression levels of circCDK13 were extracted from the microarray data (*n* = 5 biologically independent samples). **c** The results of RT-qPCR analysis showed that the expression of circCDK13 was down-regulated in the wound of diabetic mice (*n* = 8 independent mice). WWs, Normal wild-type mice wounds; DWs, diabetic mice wounds. **d, e** Relative expression levels of circCDK13 in HDFs and HEKs treated with different concentrations of AGE-BSA (*n* = 3 biologically independent samples). **f** Schematic illustration showing the circularization of CDK13 exon 2 forming circCDK13. Sanger sequencing following PCR was used to show the "head-to-tail" splicing of circCDK13. **g** The presence of circCDK13 was validated in HDFs and HEKs by RT-qPCR. Divergent primers amplified circCDK13 in cDNA but not in genomic DNA. GAPDH was used as a negative control. **h, i** Actinomycin D treatment was used to evaluate the stability of circCDK13 and CDK13 mRNA in HDFs and HEKs (*n* = 3 biologically independent samples). **j, k** Resistance of circCDK13 and CDK13 mRNA to digestion with RNase R exonuclease was detected by RT-qPCR (*n* = 3 biologically independent samples). RNase R exonuclease specifically degraded linear RNAs but not circRNAs. **l** Localization of circCDK13 detected by fluorescence in situ hybridization (FISH) assay in HDFs and HEKs. Scale bar = 50 μm. **m, n** Nuclear and cytoplasmic fractionation revealed the level of circCDK13 in the nucleus and cytoplasm of HDFs and HEKs (*n* = 3 independent experiments). U6 and GAPDH were used as positive controls in the nucleus and cytoplasm, respectively. In (**g, l**) three independent experiments were performed and similar results were obtained. Comparisons were performed by two-tailed Student's *t* test in (**b, c, j, k**) and one-way ANOVA followed by Tukey's multiple comparisons test in (**d, e**). Data are presented as mean values ± SD. Source data are provided as a Source Data file.

area of circCDK13 knockdown group was remarkably lower than that of its control group and the migration area of circCDK13 overexpression group was higher than that of corresponding control group (Fig. 2m, n; Supplementary Fig. 6g–j). Additionally, the results of transwell assay demonstrated that circCDK13 knockdown decreased and circCDK13 overexpression increased the number of migrated cells (Fig. 2o, p; Supplementary Fig. 6k–n). Moreover, we detected the CDK13 mRNA (Supplementary Fig. 7a–d) and protein expressions (Supplementary Fig. 7e) and found that they were unaffected by siRNAs and over-expression vector of circCDK13. The results demonstrated that siRNAs used in our study didn't degrade CDK13 linear mRNA transcript and

circCDK13 couldn't increase expression of CDK13 mRNA and protein by a positive feedback pathway reported before[18]. Collectively, the above findings indicated that circCDK13 rather than CDK13 protein played an essential role in regulating the proliferation and migration of HDFs and HEKs.

## circCDK13 interacts with IGF2BP3 in an m⁶A-dependent manner

Recently, studies have found that some cytoplasm-retained circRNAs could interact with RNA-binding proteins (RBPs) and participate in regulating cell biological functions[19–21]. To further elucidate the molecular mechanism by which circCDK13 promotes the

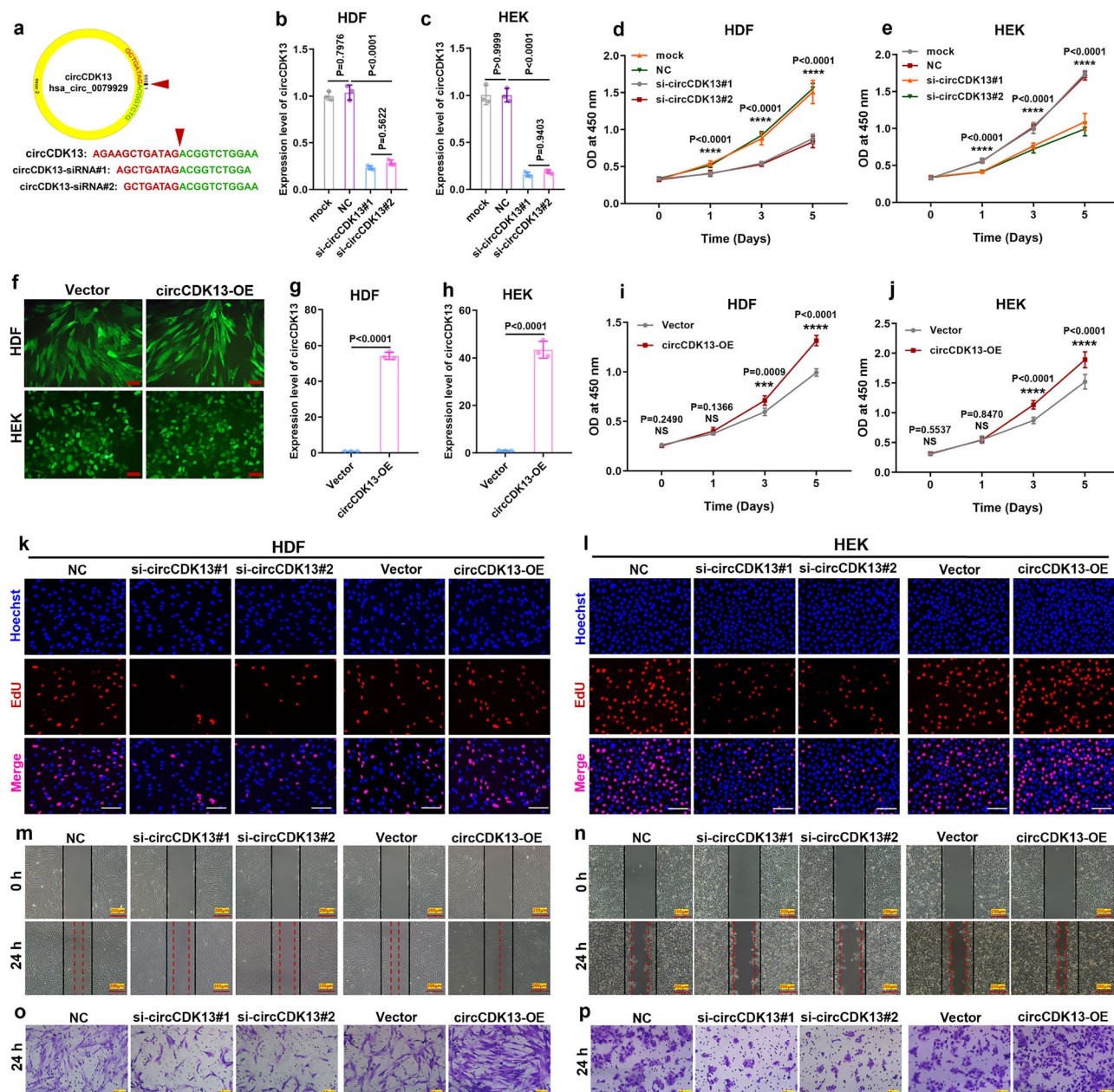

**Fig. 2 | circCDK13 promotes the proliferation and migration of HDFs and HEKs in vitro. a** Schematic representation and target sequences of the siRNAs specific to the back splice junction of circCDK13. **b, c** RT-qPCR results showed that two siRNAs targeting the back-splicing site of circCDK13 significantly decreased the expression levels of circCDK13 in HDFs and HEKs (*n* = 3 biologically independent samples). **d, e** CCK-8 assay showed that knockdown of circCDK13 induced the repression of viability of HDFs and HEKs (*n* = 6 biologically independent samples). In the comparison between si-circCDK13#1 (or si-circCDK13#2) group and NC (or mock) group, **** represents *P* < 0.0001. **f** HDFs and HEKs expressed green fluorescent protein (GFP), indicating that pLC5-ciR plasma was successfully introduced into cells and expressed stably. Scale bar, 100 μm. **g, h** RT-qPCR was performed to detect circCDK13 after overexpression of circCDK13 in HDFs and HEKs (*n* = 3 biologically independent samples). **i, j** CCK-8 assay showed that overexpression of circCDK13 upregulated viability of HDFs and HEKs (*n* = 6 biologically independent samples). **k, l** EdU incorporation assay was performed to assess DNA synthesis in HDFs and HEKs. Red fluorescence represented EdU-positive cells, while blue fluorescence represented total cells. Scale bar, 100 μm. **m, n** Wound healing assay was performed to measure the motility of HDFs and HEKs. Scale bar, 200 μm. **o, p** Representative images of transwell migration assay of HDFs and HEKs. Scale bar, 50 μm. In (**k–p**) five independent experiments were performed and similar results were obtained. Comparisons were performed by one-way ANOVA followed by Tukey's multiple comparisons test in (**b–e**) and two-tailed Student's *t* test in (**g–j**). Data are presented as mean values ± SD. Source data are provided as a Source Data file.

proliferation and migration of HDFs and HEKs, we performed an RNA pull-down assay (Fig. 3a), followed by electrophoresis, silver staining, and liquid chromatography-tandem mass spectrometry (LC-MS/MS) assays. As shown in Fig. 3b, we observed a differential band between 60 KD and 75 KD. Subsequently, 94 specific RBPs in the pull-down product of the circCDK13 probe group were identified by LC-MS/MS (Fig. 3c, Supplementary Fig. 8). In addition, 37 RBPs

that have binding sites with circCDK13 were predicted by catRAPID omics v2.0 (Fig. 3d). Subsequently, we conducted an intersection analysis of our LC-MS/MS data with the aforementioned 37 RBPs, leading us to identify IGF2BP3 and PTBP1 as the probable candidate RBPs that interact with circCDK13 (Fig. 3d). Finally, IGF2BP3 was identified as a potential RBP that interacts with circCDK13, while PTBP1 was eliminated as a candidate due to its molecular weight

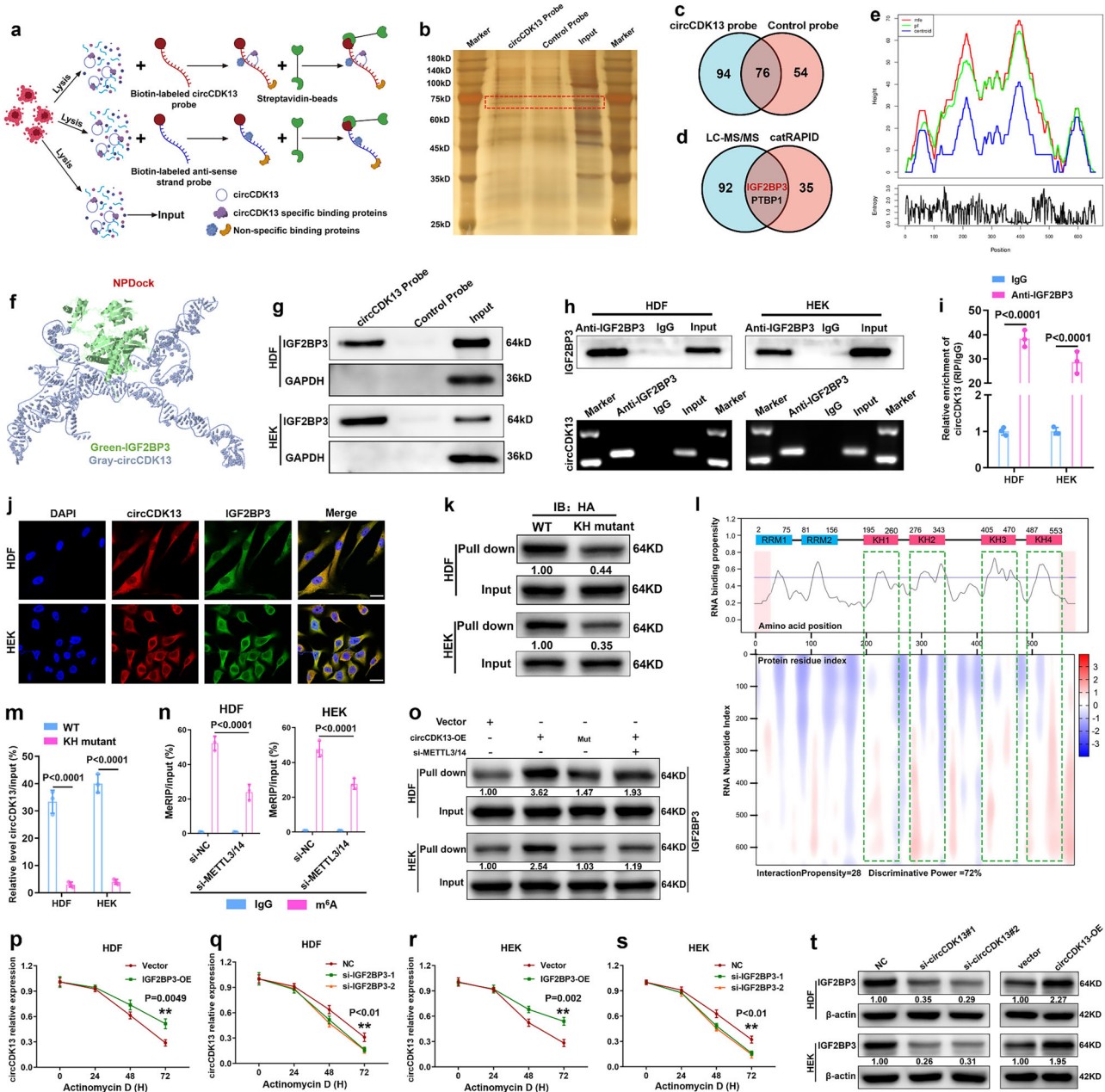

**Fig. 3 | circCDK13 interacts with IGF2BP3 in an m⁶A-dependent manner.**
**a** Schematic of RNA pull-down assay. **b** Silver staining of proteins pulled down by biotin-labeled probe specific for circCDK13 and control probe. **c** Venn diagram of RNA pull-down protein products after LC-MS/MS analysis. **d** Venn diagram of the overlapping RBPs between circCDK13-interacting proteins identified by LC-MS/MS and RBPs predicted to bind to circCDK13 by catRAPID omics v2.0. **e** Mountain plot representing the minimum free energy (MFE, red), the thermodynamic ensemble (green) and the centroid structures (blue) of circCDK13. **f** Graphical representation of the molecular docking between circCDK13 and IGF2BP3 protein using NPDock. **g** The interaction between circCDK13 and IGF2BP3 was verified by RNA pull-down and western blot assays. GAPDH was used as a negative control. **h** RIP assay validated the interaction between circCDK13 and IGF2BP3. **i** RT-qPCR analysis of relative enrichment of circCDK13 in IGF2BP3 RIP products. **j** FISH and IF co-staining showing co-localization of circCDK13 (red) with IGFB2P3 (green) in HDFs and HEKs. Scale bar, 50 μm. **k** RNA pull-down and western blot assays confirmed that IGF2BP3 bound to circCDK13 through the KH domains. **l** Prediction of circCDK13-IGF2BP3

interaction using the catPAPID algorithm and schematic of IGF2BP3 with functional protein domains. **m** Analysis for the circCDK13 enrichment, relative to input. RIP assay was performed using HA antibody in HDFs and HEKs. **n** Gene-specific m⁶A RT-qPCR assay detected m⁶A modification of circCDK13 in HDFs and HEKs with or without METTL3/14 knockdown. **o** RNA pull-down and western blot assays were used to detect the interaction between circCDK13 and IGF2BP3 in the indicated group. **p–s** The half-life of circCDK13 after treatment with 5 μg/L actinomycin D for the indicated times in HDFs and HEKs with knockdown or overexpression of IGF2BP3. **t** The protein level of IGF2BP3 in HDFs and HEKs with knockdown or overexpression of circCDK13 treated with cycloheximide (25 μg/ml) for 12 h. In (**b**, **g**, **h**, **k**, **o**, **t**) three independent experiments were performed and similar results were obtained. Comparisons were performed by one-way ANOVA followed by Dunnett's multiple comparisons test in (**q**, **s**) and two-tailed Student's *t* test in (**i**, **m**, **n**, **p**, **r**). Data are presented as mean values ± SD (*n* = 3 biologically independent samples). Source data are provided as a Source Data file.

falling outside the range of 60 KD to 75 KD. The secondary structure of circCDK13 with the minimum free energy (MFE) was predicted by utilizing the online tool RNAfold (Fig. 3e, Supplementary Fig. 9a). Next, NPDock was used to predict the physical interaction between circCDK13 and IGF2BP3 (Fig. 3f). The binding between circCDK13 and IGF2BP3 was further confirmed by RNA pull-down and RIP-qPCR assays (Fig. 3g–i). Moreover, we carried out an RNA FISH-immunofluorescence (FISH-IF) analysis and found that circCDK13 colocalized with IGF2BP3 in the cytoplasm of HDFs and HEKs (Fig. 3j). These data suggest that circCDK13 physically interacted with IGF2BP3 in HDFs and HEKs.

CatRAPID was used to further explore the binding site between IGF2BP3 and circCDK13, and the results showed that IGF2BP3 contained two RNA recognition motifs (RRMs) and four K homology (KH) domains, and might directly interact with circCDK13 through KH domains (Fig. 3l). Several studies have demonstrated that KH domains of IGF2BP3 were indispensable for the interaction between IGF2BP3 and RNAs[22–25]. To investigate the essentiality of the KH domains in the interaction between IGF2BP3 and circCDK13, we constructed a mutant IGF2BP3 with GxxG to GEEG mutation in the KH domains. The results indicated that mutation of the KH domains remarkably impaired the interaction between IGF2BP3 and circCDK13 (Fig. 3k, m). Next, the RBP suite online website was utilized to predict the binding site between circCDK13 and IGF2BP3, revealing that IGF2BP3 could bind to the "ACAAACA" motif (Supplementary Fig. 9b). Analysis of the nucleotide sequence of circCDK13 by catRAPID fragment revealed that nucleotides within the 280−430 bp range possess a high potential for binding with IGF2BP3 (Supplementary Fig. 9c). Furthermore, a potential IGF2BP3-binding region ("ACAAACA", highlighted in yellow) was identified within circCDK13 (Supplementary Fig. 9d, up panel). Subsequently, we conducted an analysis of the circCDK13 mutant harboring mutations from "ACAAACA" to "UGUUUGU" (highlighted in green) (Supplementary Fig. 9d, down panel), and found that mutation in the circCDK13 significantly decreased the interaction between IGF2BP3 and circCDK13 (Fig. 3o). In addition, circCDK13 was found to have m[6]A modification sites, analyzed by the circPrimer software and the circBank online platform (Supplementary Fig. 9e–g). Studies have shown that IGF2BP3 acted as an m[6]A-binding protein, and reducing m[6]A levels could impair the capability of IGF2BP3 to bind to its target RNAs[22,25]. The m[6]A modification has been proposed to be installed by a multicomponent methyltransferase complex (MTC), which consists of a core component methyltransferase-like 3 (METTL3) and methyltransferase-like 14 (METTL14) heterodimer[20]. To investigate whether interactions between circCDK13 and IGF2BP3 are regulated via an m[6]A-dependent manner, we conducted m[6]A RNA immunoprecipitation (MeRIP) of circCDK13, and the significant enrichment of m[6]A in circCDK13 was observed (Fig. 3n). Moreover, the depletion of METTL3 and METTL14 reduced the level of m[6]A modification in circCDK13 (Fig. 3n) and impaired the interaction between circCDK13 and IGF2BP3 (Fig. 3o), while leaving the level of circCDK13 unaltered (Supplementary Fig. 10a, b). The above research results demonstrated that the KH domains of IGF2BP3, the "ACAAACA" sequence and m[6]A modification of circCDK13 were essential for the interaction between circCDK13 and IGF2BP3.

## The interaction between circCDK13 and IGF2BP3 enhances the stability of each other

Previous studies confirmed that IGF2BP3 could increase the stability of RNAs binding to it[26]. Additionally, circRNAs could also regulate the stability of RBPs[25,27]. Hence, we performed qRT-PCR and western blot assays to probe the molecular consequences of the interaction of circCDK13 and IGF2BP3. To silence or upregulate IGF2BP3, IGF2BP3 siRNAs and an overexpression vector were constructed (Supplementary Fig. 11a−c). We found that IGF2BP3 knockdown decreased circCDK13 abundance, whereas IGF2BP3 overexpression

increased circCDK13 abundance (Supplementary Fig. 11d, e). To further illuminate whether IGF2BP3 could promote the production or inhibit the degradation of circCDK13, we detected the abundance of circCDK13 by silencing or overexpressing IGF2BP3 under the administration of actinomycin D, which could block the synthesis of new RNAs. Our data demonstrated that IGF2BP3 increased the abundance of circCDK13 by increasing the stability of circCDK13 (Fig. 3p–s). Furthermore, we found that when cycloheximide was used to inhibit the synthesis of new proteins, IGF2BP3 protein levels were downregulated by circCDK13 silencing or upregulated by circCDK13 overexpression (Fig. 3t), while circCDK13 failed to alter the expression level of IGF2BP3 mRNA (Supplementary Fig. 11f, g). These results indicated that the interaction between circCDK13 and IGF2BP3 could enhance each other's stability.

## circCDK13 cooperates with IGF2BP3 to promote the proliferation and migration of HDFs and HEKs

Several studies have shown that the combination of non-coding RNAs and RBPs could play a synergistic effect on the regulation of downstream target genes[19,28]. To further elucidate the role of circCDK13 and IGF2BP3 in HDFs and HEKs, serials of rescue experiments were performed. Strikingly, IGF2BP3 knockdown abolished the induction of cell proliferation and migration elicited by the circCDK13 overexpression (Fig. 4a−f, Supplementary Fig. 12a, c). Conversely, circCDK13 knockdown abrogated the promoting effects of IGF2BP3 overexpression on cell proliferation and migration (Fig. 4a−f, Supplementary Fig. 12b, d). These results indicated that circCDK13 and IGF2BP3 might exert a synergistic effect on favoring the proliferation and migration of HDFs and HEKs. Studies reported that IGF2BP3 could increase the stability of downstream target mRNAs, such as CD44 and c-MYC[26,29]. We observed that the knockdown of endogenous IGF2BP3 significantly reduced the protein levels of c-MYC, CD44, and CyclinD1, while upregulation of IGF2BP3 dramatically increased the protein levels of c-MYC, CD44, and cyclinD1 (Supplementary Fig. 13). Afterward, administration of circCDK13 siRNAs significantly decreased the protein levels of c-MYC, CD44, and CyclinD1, while the opposite effects were found in circCDK13 overexpression group (Supplementary Fig. 14). Furthermore, we found that IGF2BP3 knockdown abolished the effects of circCDK13 overexpression on the mRNA levels of c-MYC and CD44, and circCDK13 knockdown abrogated the effect of IGF2BP3 overexpression on the mRNA levels of c-MYC and CD44 (Fig. 4g, h), indicating that the effects of circCDK13 and IGF2BP3 on c-MYC and CD44 mRNA were likely interdependent. To further prove that circCDK13 and IGF2BP3 synergistically promote the expression of c-MYC and CD44 by enhancing the stability of c-MYC and CD44 mRNA, we performed the PCR analysis of pull-down products precipitated by circCDK13 probe or IGF2BP3 antibody. As expected, we found that there was an enrichment of c-MYC and CD44 mRNA in the pull-down products (Supplementary Fig. 15), which provided the possibility for circCDK13 and IGF2BP3 to bind with c-MYC and CD44 mRNA for stabilization. However, it's unclear whether circCDK13 or IGF2BP3 directly binds to c-MYC and CD44 mRNA and whether other mediators are involved in the binding of circCDK13-IGF2BP3 complex to c-MYC and CD44 mRNA. One study reported that IGF2BP3 could increase c-MYC mRNA stability by binding to the coding region instability determinant (CRD) residing in the 3′-terminus of the c-MYC coding region[26]. In another study, IGF2BP3 was found to bind to the 3′-UTR of CD44 mRNA for its stabilization[29]. Combining literature reports and our results, we speculated that circCDK13 and IGF2BP3 synergistically promoted the expression of c-MYC and CD44 by enhancing the stability of c-MYC and CD44 mRNA in HDFs and HEKs. Furthermore, we observed significantly lower expression levels of IGF2BP3, c-MYC and CD44 proteins in epidermal keratinocytes and dermal fibroblasts from the DFU group compared with the

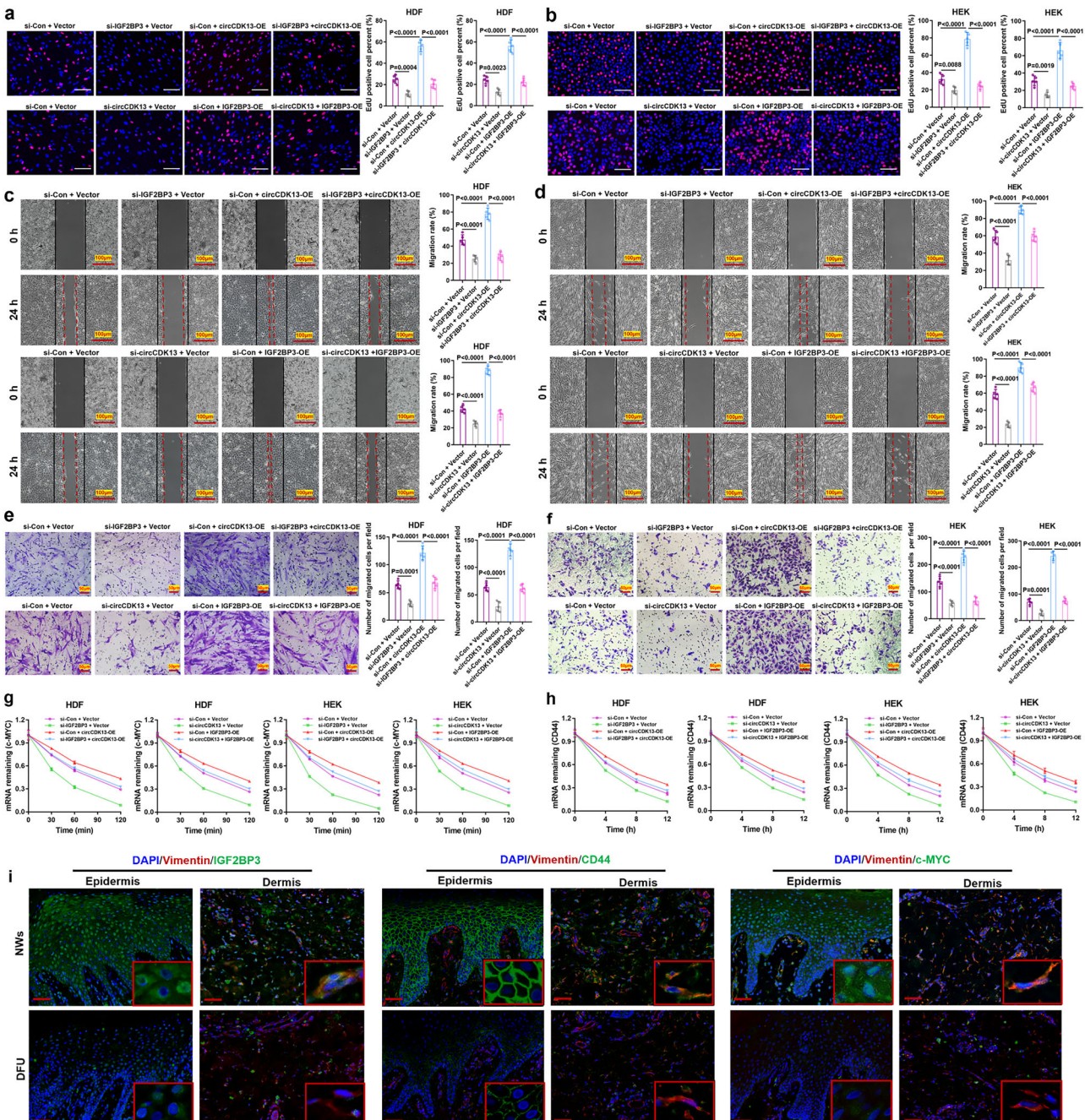

**Fig. 4 | circCDK13 cooperates with IGF2BP3 to promote the proliferation and migration of HDFs and HEKs. a**, **b** In the IGF2BP3-knockdown HDFs and HEKs with ectopically expressed circCDK13 and in the circCDK13-knockdown HDFs and HEKs with ectopically expressed IGF2BP3, EdU incorporation assay was performed to assess DNA synthesis. Red fluorescence represented EdU-positive cells, while blue fluorescence represented total cells. Quantification histogram represented EdU positive cell percentage (*n* = 5 biologically independent samples). Scale bar, 100 μm. **c**, **d** Representative images of wound healing assay of HDFs and HEKs. Quantification histogram represented migration rate (*n* = 5 biologically independent samples). Scale bar, 100 μm. **e**, **f** Representative images of transwell migration assay of HDFs and HEKs. Quantification histogram represented the number of migrated cells (*n* = 5 biologically independent samples). Scale bar, 50 μm. **g**, **h** The half-life of c-MYC and CD44 after treatment with 5 μg/mL actinomycin D for the indicated times (*n* = 3 biologically independent samples). **i** Immunofluorescent staining of IGF2BP3, CD44, and c-MYC in epidermal keratinocytes and dermal fibroblasts from DFU and NWs. Scale bar, 50 μm. DAPI staining was used to label the nuclei (blue). Vimentin staining was used to label HDFs (red). Three independent experiments were performed and similar results were obtained. Comparisons were performed by one-way ANOVA followed by Tukey's multiple comparisons test in (**a**–**f**). Data are represented as mean ± SD. Source data are provided as a Source Data file.

NWs group (Fig. 4i). Based on the above experimental results, we inferred that circCDK13 and IGF2BP3 synergistically enhance the stability of c-MYC and CD44 mRNA, increase c-MYC and CD44 protein levels, and thereby enhance the proliferation and migration of HDFs and HEKs.

## Preparation and characterization of engineered sEVs bearing circCDK13

To improve the potential application of circCDK13 in wound healing and tissue regeneration, engineered sEVs containing a higher abundance of circCDK13 were constructed. We used circCDK13 lentivirus or

vector to infect human placental chorionic plate-derived MSCs (CP-MSCs), then sEVs were collected by ultracentrifugation (Supplementary Fig. 16a). The results of PCR analysis suggested that the over-expression of circCDK13 was successfully achieved in CP-MSCs (Fig. 5a), and the expression level of CDK13 mRNA in CP-MSCs and their secreted sEVs fail to change (Supplementary Fig. 16b, c). Afterward, isolated sEVs were verified by transmission electron microscopy (TEM), nanoparticle tracking analysis (NTA), and western blot assay. These engineered sEVs exhibited a typical "saucer-like" structure when observed under TEM (Fig. 5b), with size peaking at 114 nm in diameter as determined by NTA (Fig. 5c). Western blot analysis of characteristic sEVs membrane proteins, CD63, CD9 TSG101, and Alix further confirmed the identity of the sEVs (Fig. 5d). circCDK13 overexpressed in CP-MSCs was successfully incorporated into the CP-MSC-derived sEVs as evidenced by the fact that circCDK13 expression was dramatically increased in the circCDK13$^{OE}$-sEVs group compared to the N-sEVs group (Fig. 5e). Furthermore, our findings indicated that circCDK13 is protected within intact sEVs as evidenced by the fact that treatment of circCDK13$^{OE}$-sEVs with RNase A/T1 mix did not reduce the level of circCDK13, whereas co-treatment of circCDK13$^{OE}$-sEVs with 1% Triton X-100, which could degrade sEVs membranes, induced a decrease of circCDK13 within the sEVs (Fig. 5f). The above results indicated that the engineered sEVs overexpressing circCDK13 were successfully prepared.

### circCDK13$^{OE}$-sEVs promote the proliferation and migration of HDFs and HEKs in vitro

Numerous studies have confirmed that AGEs were one of the important causes of poor wound healing in diabetes[3,30,31]. In the present study, we have validated that AGE-BSA could inhibit the proliferation and migration of HDFs and HEKs. To explore whether circCDK13$^{OE}$-sEVs could antagonize the inhibition effect of AGE-BSA, firstly we determined whether circCDK13$^{OE}$-sEVs could be internalized into HDFs and HEKs. After co-incubating CM-DiI labeled sEVs with HDFs and HEKs for 6 h, we observed that HDFs and HEKs successfully internalized sEVs (Fig. 5g). Furthermore, we observed that N-sEVs could partially abrogated the inhibitory effect of AGE-BSA on the proliferation and migration of HDFs and HEKs, and circCDK13$^{OE}$-sEVs exhibited a stronger ability to promote cell proliferation and migration compared to N-sEVs (Fig. 5i–n). The results of PCR analysis showed that the abundance of circCDK13 was dramatically increased in cells co-cultured with circCDK13$^{OE}$-sEVs (Fig. 5h). In addition, it was observed that the protein expression levels of IGF2BP3, c-MYC, CD44, and Cyclin D1 were significantly augmented in HDFs and HEKs upon co-incubation with circCDK13$^{OE}$-sEVs (Fig. 5o).

The hyperglycemic microenvironment is another negative factor associated with diabetic wound healing. Therefore, HDFs were treated with high-concentration D-glucose (HG) to simulate the hyperglycemic microenvironment in diabetic wounds. We observed that when the concentration of D-glucose reached 35 mM, the proliferation and migration of HDFs were inhibited (Supplementary Fig. 17). Next, HDFs were cultured in 35 mM D-glucose complete medium and then co-incubated with N-sEVs or circCDK13$^{OE}$-sEVs. We found that N-sEVs and circCDK13$^{OE}$-sEVs could abolish the inhibitory effect of HG on the proliferation and migration of HDFs and HEKs, and that circCDK13$^{OE}$-sEVs performed better than N-sEVs in promoting the proliferation and migration of HDFs (Supplementary Fig. 18a–g). The results of PCR analysis demonstrated that the abundance of circCDK13 was reduced in HDFs treated with HG, while the abundance of circCDK13 was increased in HDFs co-incubated with circCDK13$^{OE}$-sEVs (Supplementary Fig. 18h). Furthermore, we observed that the protein expression levels of IGF2BP3, c-MYC, CD44, and Cyclin D1 were diminished in HDFs treated with HG (Supplementary Fig. 18i). Conversely, when HDFs were treated with circCDK13$^{OE}$-sEVs, the protein expression levels of IGF2BP3, c-MYC,

CD44, and Cyclin D1 were obviously increased (Supplementary Fig. 18i). Taken together, these findings preliminarily suggested that circCDK13$^{OE}$-sEVs antagonize the inhibition effect of AGE-BSA or HG on the proliferation and migration of HDFs and HEKs partly by mediating the circCDK13-IGF2BP3-CD44/c-MYC signaling pathway.

### circCDK13$^{OE}$-sEVs accelerate wound healing in vivo

To investigate whether circCDK13$^{OE}$-sEVs can accelerate diabetic wound healing in vivo, two full-thickness cutaneous wounds were created on the back of each db/db diabetic mice, followed by sub-cutaneous injection of circCDK13$^{OE}$-sEVs, N-sEVs, or an equal volume of PBS (Fig. 6a). No death or abnormality was observed in any animal during the postoperative period. Digital photographs of wounds showed that much faster wound closure was found when exposed to circCDK13$^{OE}$-sEVs, as determined by smaller wound areas measured on days 3, 7, 14, and 21 post-wounding compared with N-sEVs and Con groups (Fig. 6b, c, g). Subsequently, H&E staining was conducted to evaluate the wound length in each experimental group, and the results were consistent with the previously mentioned wound area measurements (Fig. 6d, h). Re-epithelialization is critical for wound healing. On days 3 and 7, we evaluated the epithelial tongue length by H&E staining, and found that circCDK13$^{OE}$-sEVs exhibited a stronger ability to promote re-epithelialization than N-sEVs (Fig. 6e, i). More encouragingly, mature skin structures such as hair follicles and sebaceous glands were observed in circCDK13$^{OE}$-sEVs- treated diabetic wounds on day 21 (Fig. 6e). Histological analysis of Masson's trichrome-stained sections indicated that on days 14 and 21 post-wounding, the collagen fibers in the circCDK13$^{OE}$-sEVs-treated diabetic wounds were thicker and more regular compared with N-sEVs group (Fig. 6f). Furthermore, the results of western blot and immunofluorescence assays showed that the expression levels of IGF2BP3, c-MYC, and CD44 in the circCDK13$^{OE}$-sEVs group were higher than that in the N-sEVs group (Fig. 6j–l).

To further explore the efficacy of circCDK13$^{OE}$-sEVs, type I diabetic rats were established by intraperitoneal injection of STZ, and two full-thickness cutaneous wounds were created on the back per rat (Fig. 7a). Through gross wound evaluation, H&E staining, and Masson's trichrome-stained analysis, it was once again confirmed that compared to N-sEVs, circCDK13$^{OE}$-sEVs had more significant advantages in promoting wound re-epithelialization, granulation tissue formation and regeneration of skin appendages (Fig. 7b–i). Moreover, compared with the N-sEVs group, the expression levels of IGF2BP3, c-MYC, and CD44 in the wound tissue of the circCDK13$^{OE}$-sEVs group were higher (Fig. 7j–l).

In summary, compared with N-sEVs, circCDK13$^{OE}$-sEVs performed better in promoting diabetic wound healing, which is mainly manifested by enhancing wound re-epithelialization, granulation tissue formation, collagen deposition, and the regeneration of hair follicles and sebaceous glands. Mechanistically, circCDK13$^{OE}$-sEVs might promote the healing of diabetic wounds through the formation of circCDK13-IGF2BP3-CD44/c-MYC ternary complex to promote the expression of CD44 and c-MYC.

## Discussion

In the present study, we revealed that impaired healing of diabetic wounds was tightly associated with the downregulation of circCDK13, and then confirmed a potential wound healing-promoting mechanism that circCDK13 directly interacted with IGF2BP3 to form a circRNA-protein-mRNA ternary complex, which synergistically enhanced the stability of CD44 and c-MYC mRNA to promote the proliferation and migration of HDFs and HEKs. Subsequently, we successfully constructed engineered sEVs bearing circCDK13 and corroborated that circCDK13$^{OE}$-sEVs could abolish the inhibitory effect of AGE-BSA or HG on proliferation and migration of HDFs and HEKs. Finally, we confirmed that in the wounds of

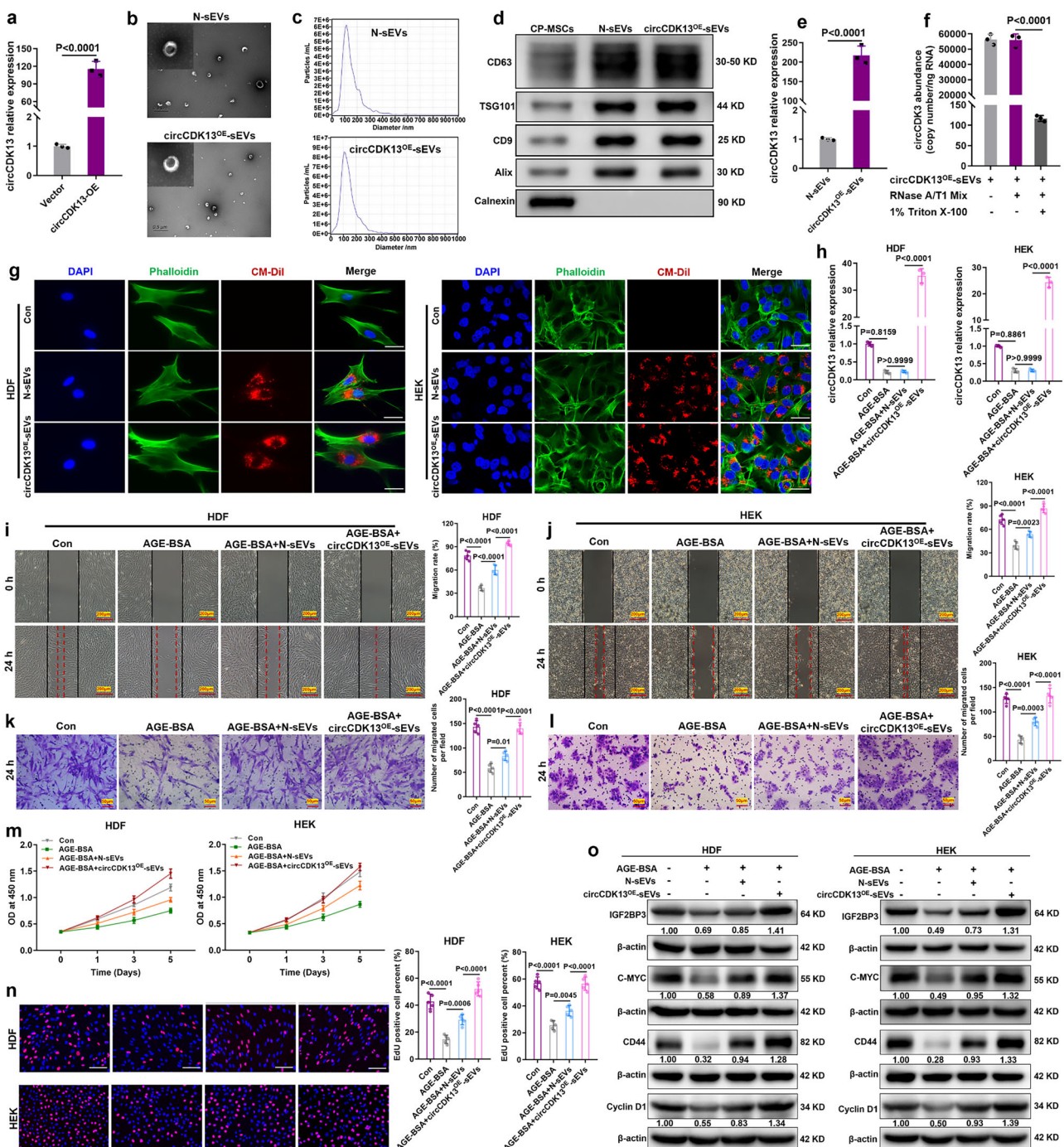

**Fig. 5 | circCDK13^OE-sEVs promote the proliferation and migration of HDFs and HEKs. a** RT-qPCR analysis of circCDK13 abundance in CP-MSCs transduced with vector or circCDK13 lentivirus (*n* = 3 biologically independent samples). **b** Representative TEM photomicrographs of N-sEVs and circCDK13^OE-sEVs. Scale bars, 500 nm. **c** The size distribution and concentration of N-sEVs and circCDK13^OE-sEVs measured by NTA. **d** CD63, TSG101, CD9, Alix, and Calnexin were detected by western blot assays in the whole-cell lysate (CP-MSCs) and purified sEVs of CP-MSCs (N-sEVs and circCDK13^OE-sEVs). **e** RT-qPCR analysis of circCDK13 abundance in N-sEVs and circCDK13^OE-sEVs (*n* = 3 biologically independent samples). **f** Absolute qPCR analysis of circCDK13 copy numbers in circCDK13^OE-sEVs after treatments with RNase A/T1 Mix and 1% Triton X-100 for 30 min (*n* = 3 biologically independent samples). **g** Internalization of sEVs by HDFs and HEKs. sEVs were marked by the red fluorescence (CM-DiI) and cytoskeleton were marked by green fluorescence (phalloidin). Scale bar, 25 μm. **h** RT-qPCR analysis of circCDK13 abundance in HDFs and HEKs after co-incubation with N-sEVs or circCDK13^OE-sEVs (sEVs concentration, 2 × 10^10 particles/ml) for 6 h (*n* = 3 biologically independent samples). **i, j** Wound

healing assay was performed to measure the motility of HDFs and HEKs. Quantification histogram represented migration rate (*n* = 5 biologically independent samples). Scale bar, 200 μm. **k, l** Representative images of transwell migration assay of HDFs and HEKs. Quantification histogram represented the number of migrated cells (*n* = 5 biologically independent samples). Scale bar, 50 μm. **m** CCK-8 assay was performed after HDFs and HEKs were incubated with N-sEVs or circCDK13^OE-sEVs (sEVs concentration, 2 × 10^10 particles/ml) at the indicated time points (*n* = 6 biologically independent samples). **n** EdU incorporation assay was performed to assess DNA synthesis in HDFs and HEKs. Quantification histogram represented EdU positive cell percentage (*n* = 5 biologically independent samples). Scale bar, 100 μm. **o** Western blots analysis of IGF2BP3, c-MYC, CD44, and cyclin D1 protein expression levels in HDFs and HEKs of different treatment groups. In (**d, g, o**) three independent experiments were performed and similar results were obtained. Comparisons were performed by one-way ANOVA followed by Tukey's multiple comparisons test in (**f, h–l, n**) and two-tailed Student's *t* test in (**a, e**). Data are presented as mean values ±SD. Source data are provided as a Source Data file.

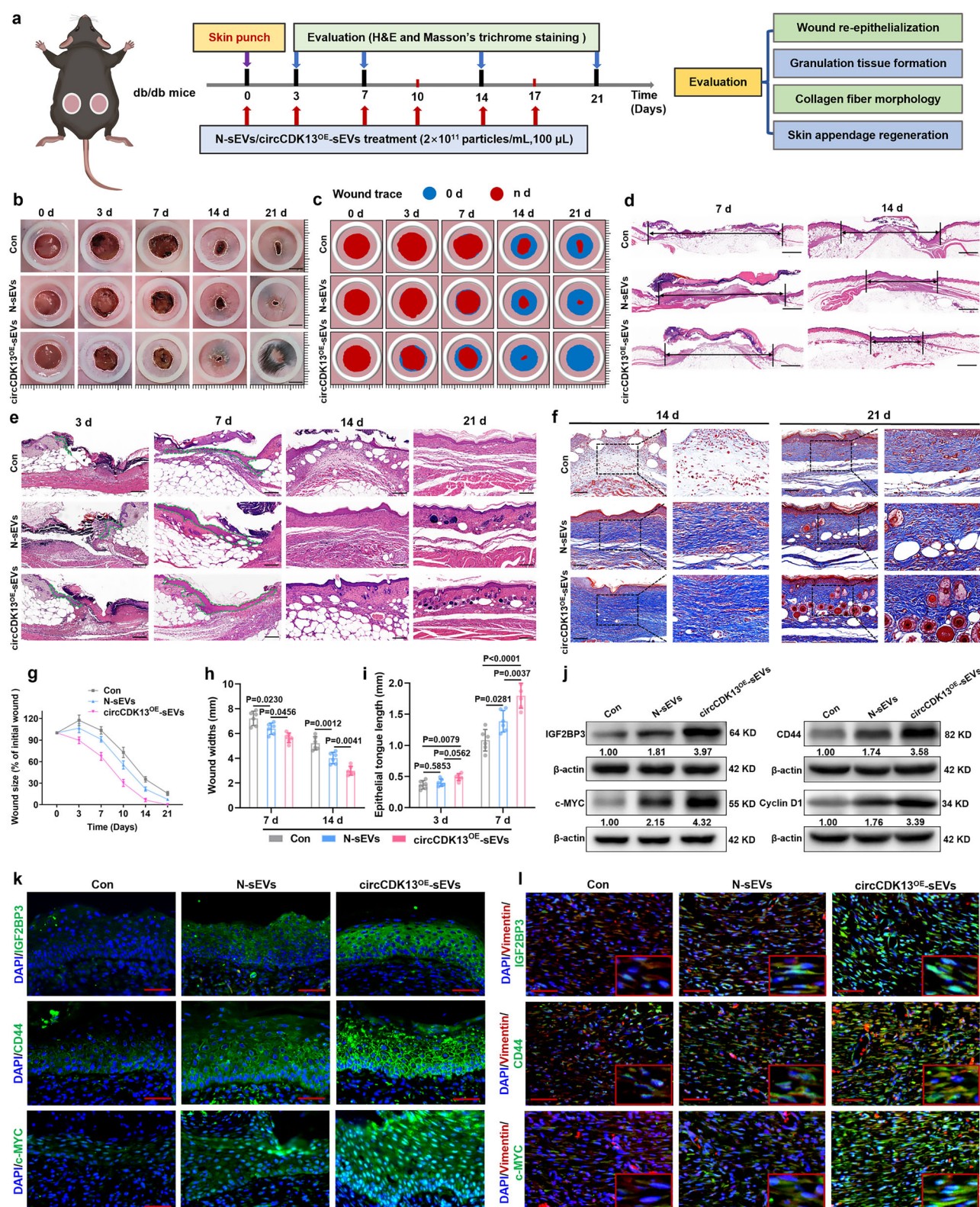

db/db diabetic mice and STZ-induced type I diabetic rats, circCDK13[OE]-sEVs could accelerate re-epithelialization and granulation tissue formation, and promote wound remodeling and regeneration of skin appendages.

circRNAs have unique advantages in the development of disease diagnosis and treatment strategies due to their stronger stability, tissue and disease specificity. Recently, accumulating studies have found that circRNAs were closely related to skin wound healing[8,32–34]. In this

study, we found that the expression of circCDK13 was down-regulated in diabetic wounds, and then confirmed that circCDK13 could promote the proliferation and migration of HDFs and HEKs in vitro. Furthermore, we observed that circCDK13 was more stable than CDK13 RNA and mainly located in the cytoplasm of HDFs and HEKs. Accumulating evidence has implicated that circRNAs residing in the cytoplasm could play a significant role in various pathological and physiological states by interacting with RBPs[5,19]. We found that circCDK13 directly

**Fig. 6 | circCDK13$^{OE}$-sEVs accelerate wound healing in db/db diabetic mice.**
**a** Schematic diagram of skin wound, sEVs subcutaneous injection, and skin wound harvest timeline in db/db diabetic mice. **b** Representative images of the wound area by different treatments on days 0, 3, 7, 14, and 21 after operation. **c** Simulation plots of the wound closure areas. **d** Representative images of H&E staining of wound sections. Scale bar, 1 mm. **e** High-magnification images showed wound re-epithelialization, granulation tissue formation, and regeneration of skin appendages (hair follicles and sebaceous glands). Scale bar, 200 µm. **f** Representative images of Masson's trichrome stating of wound sections. Scale bar, 200 µm. **g** Quantitative evaluation of the wound closure rate ($n = 6$ biologically independent samples). **h** Quantitative analysis of wound width ($n = 6$ biologically independent samples). **i** Quantitative analysis of epithelial tongue length ($n = 6$ biologically independent samples). **j** Western blot analysis of IGF2BP3, c-MYC, CD44, and cyclin D1 protein expression levels in skin wound tissue of different treatment groups. **k** Immunofluorescent staining of IGF2BP3, CD44, and c-MYC in epidermal keratinocytes of skin wounds. Scale bar, 50 µm. **l** Immunofluorescent staining of IGF2BP3, CD44, and c-MYC in dermal fibroblasts of skin wounds. Scale bar, 50 µm. DAPI staining was used to label the nuclei (blue). Vimentin staining was used to label HDFs (red). In (**j**, **k**, **l**) three independent experiments were performed and similar results were obtained. Comparisons were performed by one-way ANOVA followed by Tukey's multiple comparisons test in (**h**, **i**). Data are presented as mean values ± SD. Source data are provided as a Source Data file.

interacted with IGF2BP3. IGF2BP3 is a mammalian IGF2 mRNA-binding protein family member and contains 2 RNA recognition motifs (RRM) in its N-terminus and 4 hnRNPK homology (KH) domains at the C-terminus, and KH domains are indispensable for the interactions between IGF2BP3 and RNAs[22,23,26]. In our research, we verified that KH domains were essential for the binding of circCDK13 to IGF2BP3. Recently, studies have identified IGF2BP3 as an m$^6$A-binding protein that could recruit cofactor proteins to stabilize m$^6$A-modified transcripts[26]. Firstly, we predicted that there would be a large number of m$^6$A modification sites in circCDK13 using the circPrimer software and the circBank online website. Subsequently, we confirmed that the m$^6$A modification of "ACAAACA" is critical for the binding of circCDK13 and IGF2BP3. Moreover, we demonstrated that the combination of IFG2BP3 and circCDK13 increased the stability of circCDK13. Several studies have shown that circRNAs could protect the RBPs bound to them from being rapidly degraded by the ubiquitin-proteasome system, and thereby enhancing the stability of RBPs[25,35]. We found that circCDK13 could increase the stability of IGF2BP3 since altering circCDK13 expression failed to affect the expression level of IGF2BP3 mRNA, while knocking down or overexpressing circCDK13 under the application of cycloheximide, which could block new protein synthesis, decreased or increased the protein level of IGF2BP3.

circRNA-protein-mRNA ternary complexes are frequently observed in circRNA-protein interactions, wherein circRNAs facilitate the binding of RBPs to mRNAs, thereby enhancing mRNAs stability and translation[19]. For instance, circNSUN2 forms a ternary complex with IGF2BP2 and high mobility group A (HMGA2) mRNA in the cytoplasm which stabilizes mRNA, and then upregulation of HMGA2 induces epithelial-mesenchymal transition and enhances colorectal cancer aggressiveness[36]. In a similar manner, the formation of the circFNDC3B-IGF2BP3-CD44 mRNA ternary complex serves to stabilize mRNA and upregulates CD44, thereby facilitating the migration and invasion of gastric cancer cells[37]. Our data showed that silencing IGF2BP3 in circCDK13-overexpressing cells or knocking down circCDK13 in IGF2BP3-overexpressing cells decreased the stability of CD44 and c-MYC mRNA, and weakened the synergistic effect, induced by the interaction between IGF2BP3 and circCDK13, on promoting proliferation and migration of HDFs and HEKs.

CD44, functioning as a cell surface receptor, assumes a crucial role in facilitating cell-cell interactions, cell adhesion, and migration, thereby enabling cells to perceive and react to alterations in the microenvironment of the tissue[29,38,39]. As a transcription factor, c-MYC binds DNA in a non-specific manner, yet also specifically recognizes the core sequence 5′-CAC[GA]TG-3′, consequently initiating the transcription of genes associated with growth and engaging in diverse cellular processes such as proliferation, differentiation, apoptosis, and metabolism[40,41]. In our research, we found that knockdown of circCDK13 or IGF2BP3 decreased CD44, c-MYC, and Cyclin D1 protein levels in HDFs and HEKs. Cyclin D1 is a protein encoded by human CCND1 gene, and its main function is to promote cell proliferation. Cyclin D1 plays a critical role in promoting the cell cycle from G1 phase to S phase by binding and activating G1-specific cyclin-dependent kinase CDK4/6[42,43]. Therefore, we inferred that circCDK13 and IGF2BP3 cooperated to enhance the stability of c-MYC and CD44 mRNA, thereby increasing CD44 and c-MYC protein levels and promoting the proliferation and migration of HDFs and HEKs.

circRNAs are usually composed of hundreds to thousands of bases, which are difficult to be directly synthesized in vitro like microRNAs. In addition, circRNAs are difficult to be absorb through cell membranes due to their large molecular diameter. Therefore, some researchers attempted to carry circRNAs through lentivirus or adenovirus, and achieved good therapeutic effects in preclinical studies. For example, Yang et al. confirmed that circ-Amotl could promote wound healing by injecting circ-Amotl expression plasmid directly into the wound edge[44]. Although administering circRNAs using lentiviruses or adenovirus is efficient for preclinical studies, this circRNA delivery strategy still has some biosafety issues for clinical application. Thus, exploring a safe and effective circRNA delivery strategy is particularly important. Small extracellular vesicles (sEVs) are easy to be absorbed by cells and can protect their contents from rapid degradation, so engineered sEVs were constructed to efficiently deliver circCDK13. In the present study, we prepared circCDK13$^{OE}$-sEVs by overexpressing circCDK13 in CP-MSCs and then collecting the cell culture supernatant to isolate sEVs. We confirmed that circCDK13$^{OE}$-sEVs could abolish the inhibitory effect of AGE-BSA on the proliferation and migration of HDFs and HEKs, and also abrogate the inhibitory effect of HG on HDFs proliferation and migration. Although we successfully generated circCDK13$^{OE}$-sEVs, the precise mechanisms by which circCDK13 is transferred into sEVs remain unknown. If the mechanisms of circCDK13 being transferred into sEVs can be elucidated, it is possible to improve the efficiency of sEVs loading circCDK13 and further enhance the therapeutic effect of MSC-sEVs.

Chronic non-healing wounds occur frequently in diabetic patients, such as diabetic foot and diabetic ulcers. Recently, numerous studies have confirmed that MSCs and their derivative EVs could accelerate wound healing[17,45]. Bian et al. found that sEVs derived from human decidua-derived MSCs had regenerative and protective effects on high-glucose induced senescent fibroblasts by suppressing RAGE pathway and activating Smad pathway, thereby accelerating diabetic wound healing[46]. Wei et al. showed that sEVs derived from human umbilical cord-derived MSCs promoted wound healing in diabetes through miR-17-5p mediated enhancement of angiogenesis[47]. Chu et al. found that hypoxic pretreated MSC-derived sEVs could inhibit neutrophil extracellular traps (NETs) formation in diabetic wounds and promote wound healing by delivering miR-17-5p[48]. Furthermore, our team demonstrated that miR146a-loaded engineered exosomes released from silk fibroin patch promoted diabetic wound healing by targeting IRAK1[49]. In this study, to further evaluate the effect of circCDK13$^{OE}$-sEVs on promoting wound healing, we established a full-thickness skin excision wound model on the back of db/db diabetic mice and STZ-induced type I diabetic rats, and then injected sEVs locally subcutaneously. We observed that circCDK13$^{OE}$-sEVs performed better than N-sEVs in promoting wound re-epithelialization, granulation tissue formation and regeneration of skin appendages.

In conclusion, we explored a potential mechanism that the delayed healing of diabetic wounds was closely related to the

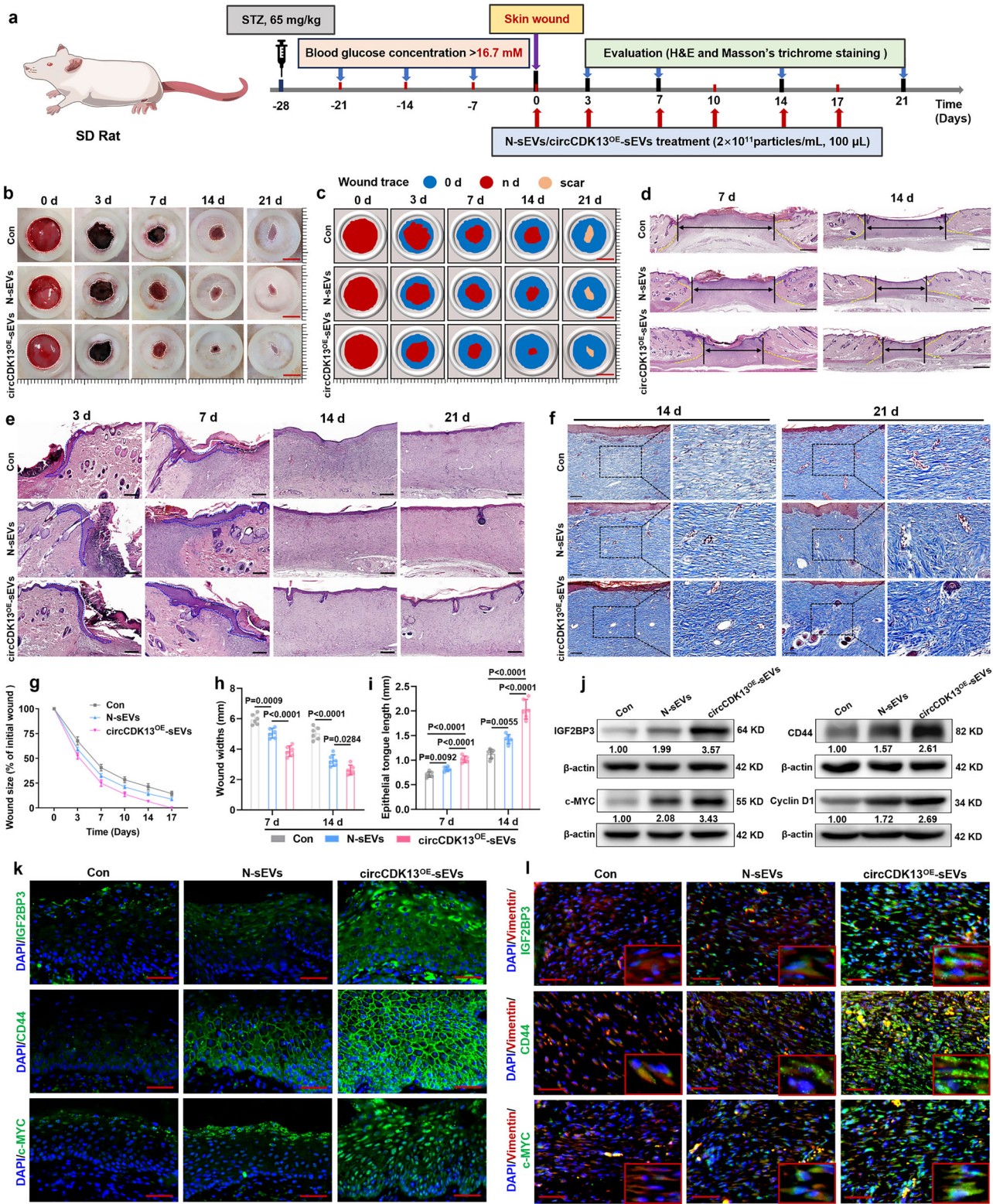

downregulation of circCDK13, and circCDK13 promoted the proliferation and migration of HDFs and HEKs through forming circCDK13-IGF2BP3-mRNA ternary complex, which could stabilize mRNA and upregulate CD44 and c-MYC (Fig. 8). In addition, we successfully constructed engineered sEVs overexpressing circCDK13, and confirmed that compared with N-sEVs, circCDK13$^{OE}$-sEVs possessed superior performance in promoting the proliferation and migration of HDFs and HEKs, as well as accelerating the healing of diabetic wounds. Our data supported that circCDK13$^{OE}$-sEVs hold promise as a promising

therapeutic strategy for diabetic wound healing with great potential for clinical applications.

## Methods

### Cell culture and tissue sample collection

The human dermal fibroblasts (HDFs) and human epidermal keratinocytes (HEKs) were purchased from the Chinese Academy of Sciences' cell bank. HDFs cultured in Dulbecco's modified Eagle's medium (DMEM, Gibco, USA) supplemented with 10% fetal bovine

**Fig. 7 | circCDK13^OE-sEVs promote wound healing in STZ-induced type I diabetic rats. a** Timeline for in vivo experiments in STZ-induced type I diabetic rats. **b** Gross view of diabetic wounds on days 0, 3, 7, 14, and 21 post-wounding with different treatments. **c** Simulation plots of the wound closure areas. **d** Representative images of H&E staining of wound sections. Scale bar, 1 mm. **e** High-magnification images showed wound re-epithelialization, granulation tissue formation, and regeneration of skin appendages (hair follicles and sebaceous glands). Scale bar, 200 μm. **f** Representative images of Masson's trichrome stating of wound sections. Scale bar, 200 μm. **g** Quantitative evaluation of the wound closure rate (*n* = 6 biologically independent samples). **h** Quantitative analysis of wound width (*n* = 6 biologically independent samples). **i** Quantitative analysis of

epithelial tongue length (*n* = 6 biologically independent samples). **j** Western blot analysis of IGF2BP3, c-MYC, CD44, and cyclin D1 protein expression levels in skin wound tissue of different treatment groups. **k** Immunofluorescent staining of IGF2BP3, CD44, and c-MYC in epidermal keratinocytes of skin wounds. Scale bar, 50 μm. **l** Immunofluorescent staining of IGF2BP3, CD44, and c-MYC in dermal fibroblasts of skin wounds. Scale bar, 50 μm. DAPI staining was used to label the nuclei (blue). Vimentin staining was used to label HDFs (red). In (**j**, **k**, **l**) three independent experiments were performed and similar results were obtained. Comparisons were performed by one-way ANOVA followed by Tukey's multiple comparisons test in (**h**, **i**). Data are presented as mean values ± SD. Source data are provided as a Source Data file.

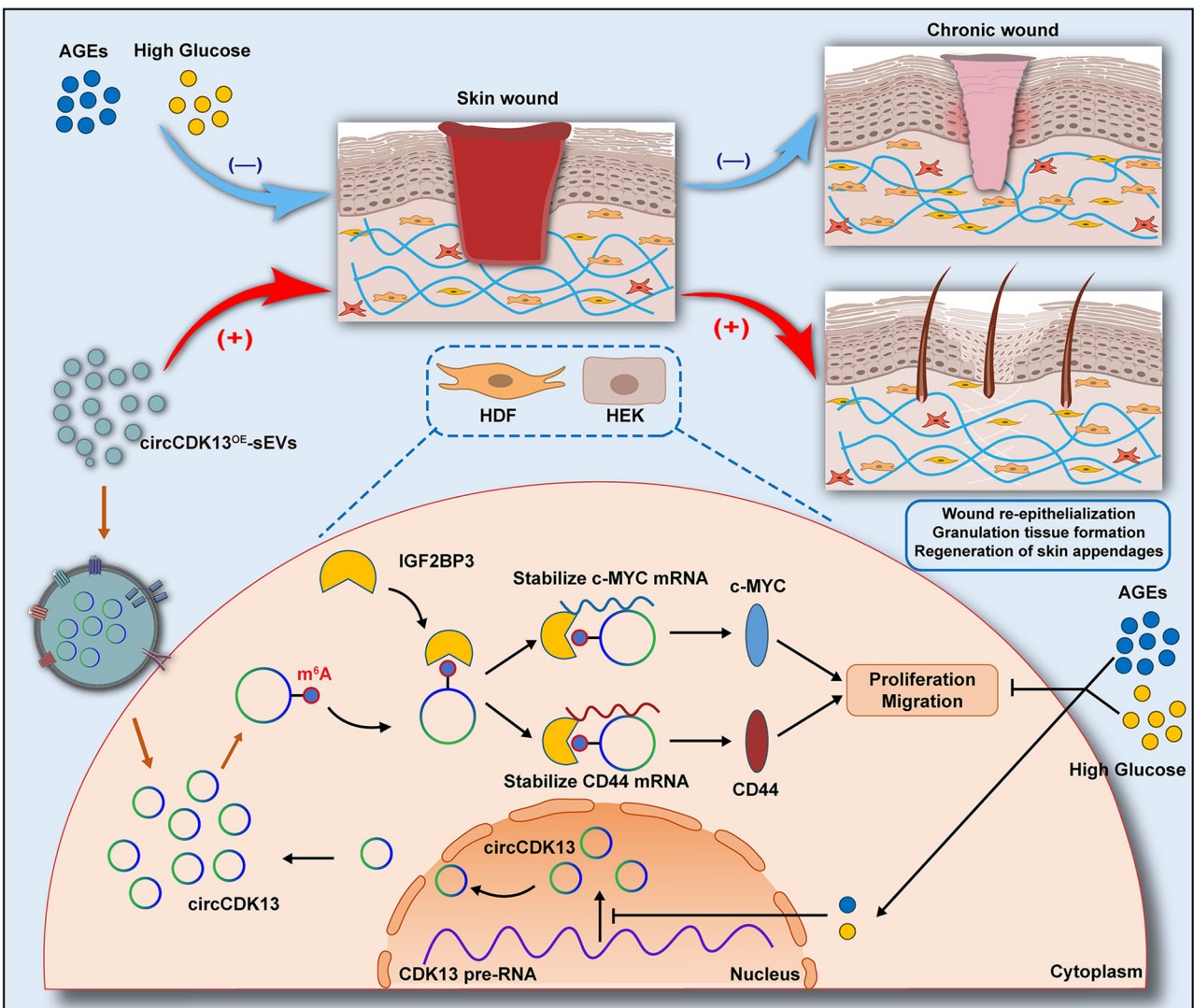

**Fig. 8 | Schematic diagram of circCDK13^OE-sEVs promoting wound healing.** AGEs accumulation and high glucose microenvironment are significant contributors to the poor healing of diabetic wounds, which could inhibit the expression of circCDK13 in skin wound repair cells. Mechanistically, circCDK13 interacts with IGF2BP3 in an m6A-dependent manner, synergistically enhancing the stability of CD44 and c-MYC mRNA, thereby boosting the expression of CD44 and c-MYC, which in turn promotes the proliferation and migration of HDFs and HEKs. Furthermore, circCDK13^OE-sEVs promote the proliferation and migration of HDFs and HEKs to accelerate wound healing partly through the circCD13-IGF2BP3-CD44/c-MYC ternary complex.

serum (FBS, Gibco, USA). HEKs cultured in EpiLife medium (Gibco, USA) supplemented with 1% human keratinocyte growth supplement (HKGS, Gibco, USA). CP-MSCs were donated by Professor Lijun Tang's team at the General Hospital of the Western Theater Command and cultured in MSC serum-free media (Yocon, China). The team's previous studies have confirmed that CP-MSCs meet the

minimal criteria of MSCs proposed by the International Society for Cellular Therapies[50]. All cells were kept in a humidified incubator (5% $CO_2$, 37 °C). All specimens from human cases and the use of CP-MSCs were obtained with the informed consent of patients and institutional approval (Ethics Committee of Chinese PLA General Hospital, project No. S2021-116-01).

## Preparation of AGE-BSA

AGE-BSA were prepared in vitro according to previously described methods[30]. In brief, 5 g/100 ml of bovine serum albumin (BSA) was co-incubated with 9 g/100 ml D-glucose and 1.5 mmol/L PMSF in sterile phosphate-buffered saline (PBS) at 37 °C for 2 months. The BSA control was incubated under similar conditions but without D-glucose. The AGE-BSA and BSA were dialyzed with a dialysis bag (8000−12000 Dalton), and then the concentration was detected using a BCA Protein Quantification Kit (Solarbio, Beijing, China). The autofluorescence intensity ($\lambda_{ex}$370 nm/$\lambda_{em}$440 nm) was measured by a fluorescence spectrometer (Spark™ Multimode Microplate Reader, TECAN, Spark 10 M) and the molecular weight change was determined by SDS-PAGE electrophoresis to identify whether the preparation of AGE-BSA was successful.

## Cell counting kit-8 (CCK-8) proliferation assay

Cells' proliferation ability was measured by Cell Counting Kit-8 (CCK-8, Dojindo Molecular Technology, Japan). Adjust the cell concentration to $2-4 \times 10^4$ cells/ml with complete medium, then add 100 μl of cell suspension to each well of a 96-well culture plate. Before the test, 10 μl of CCK-8 reagent was added to each well and incubated at 37 °C for 2 h. We then used an Spark™ Multimode Microplate Reader (Tecan, Spark 10 m) to measure the absorbance of each well at 450 nm.

## 5-Ethynyl-20-deoxyuridine (EdU) assay

EdU assays were performed to evaluate the cells' proliferation ability based on the manufacturer's instructions regarding the use of the BeyoClick™ EdU Cell Proliferation Kit with Alexa Fluor 594 (Beyotime, Shanghai, China). Cells were seeded in 24-well plates, cultured for 24 h and then incubated with 10 μM EdU solution for 12 h (HDFs) or 2 h (HEKs) in a cell incubator. After that, the cells were fixed in 4% paraformaldehyde for 30 min, permeabilized with 0.3% Triton for 10 min and subsequently stained with click reaction mixture for 30 min. Finally, the cells were stained with Hoechst 33342 for 10 min, and representative images were obtained using a Nikon A1Si Laser Scanning confocal Microscope (Nikon Instruments Inc., Japan). The cell proliferation rate was calculated using the ratio of EdU-positive cells (red) to Hoechst-positive cells (blue).

## Wound healing assay

Cells' migration ability was assessed using wound healing assay by Culture-Insert 4 Well (Ibidi, Martinsried, Germany). Cells ($2-10 \times 10^5$ cells/ml) were plated into the four wells of a culture insert located at the center of a 12-well culture plate. The culture insert was removed to reveal the wound gap when the cells reached 100% confluence. After the cells were washed with PBS to remove dissociated cellular fragments, each wound was imaged by inversion microscopy (Olympus, Japan) at 0, 12, and 24 h. Quantitative analysis was performed by measuring the gap size of the wound using the Image J software (Version 1.53k) and the results were presented as the percentage of the wound coverage.

## Transwell migration assay

Besides wound healing assay, the migration ability of HDFs and HEKs were also tested by transwell chamber (24-well culture plate, 8 μm pore size). Adjust the cell concentration to $2-4 \times 10^5$ cells/ml with serum-free medium, and then 100 μl cell suspension was added to the upper chamber of the migration well. To the contrary, 500 μL complete medium was loaded into the bottom chambers. 12 h or 24 h later, cells in the upper compartment of transwell chambers were removed with a cotton swab, then fixed with 4% paraformaldehyde for 30 min, and finally stained with crystal violet (Beyotime, Shanghai, China) for 20 min. Cells in five random separate microscope fields were photographed and counted using Image J software (Version 1.53k).

## RNA isolation and RT-qPCR

Total RNA was extracted using SteadyPure Rapid RNA Isolation Kit (Accurate Biology, China), according to the products' instructions. The RNA was quantified by measuring the absorbance at 260 nm and 280 nm using a spectrophotometer (NanoDrop Technologies, USA). Reverse transcription was performed by application of PrimeScript RT Reagent Kit (Takara, Japan). Real-time PCR was performed with SYBR® Green Pro Taq HS Premix qPCR Kit II (Accurate Biology, China) on an QuantStudio 5 Detection System (Applied Biosystems, USA). The expression of GAPDH was used as a control to calibrate the original mRNA or circRNA concentrations in tissues or cells. Target gene expression was calculated using the QuantStudio™ Design & Analysis Software v1.4.2. Detailed primer sequences can be found in Supplementary Table 1.

## RNase R treatment

To identify circular characteristics and evaluate the stability of circCDK13, total RNA (2 μg) was incubated for 15 min at 37 °C with 5 U/μg RNase R (Geneseed, Guangzhou, China) and then analyzed by RT-qPCR.

## Actinomycin D assay

HDFs ($2 \times 10^5$) or HEKs ($1 \times 10^6$) were cultured in a 6-well plate overnight and exposed to 2 μg/ml actinomycin D (Sigma, USA) for 4, 8, 12, and 24 h. At the designated time points, cells were harvested and the expression levels of circCDK13 and mRNAs were assessed using RT-qPCR.

## Nuclear and cytoplasmic extraction

A PARIS™ kit (Thermo Fisher Scientific, USA) was used to isolate nuclear and cytoplasmic fractions. HDFs and HEKs were lysed in cell fraction buffer on ice and subsequently centrifuged at 500 g for 3 min at 4 °C, and the supernatant was collected as the cytoplasmic fraction. The pelleted nuclei were incubated with cell disruption buffer and used as the nuclear fraction. Then, the RNAs of these two fractions were extracted and detected by RT-qPCR.

## RNA Fluorescence in situ hybridization (FISH) and protein immunofluorescence (IF)

Cy3-labeled probes against the back-spliced junction in circCDK13 were designed and synthesized by GenePharma (Shanghai, China). The probe sequences were listed in Supplementary Table 2. The FISH experiment was performed according to the manufacturer's instructions. For protein IF assays, cells seeded on cover glass were fixed with 4% paraformaldehyde for 30 min and washed thrice with PBS. The washed cells were permeabilized with 0.3% Triton X-100 for 20 min, blocked with 5% goat serum dissolved in PBS for 1 h, and incubated with the primary antibody (IGF2BP3, 1:100 dilution, Cat No. 14642-1-AP, ProteinTech) for overnight at 4 °C, followed by three times washes with PBS, and fluorophore-conjugated secondary antibody (Goat anti-Rabbit IgG (H + L) Cross-Adsorbed Secondary Antibody, Alexa Fluor™ 488, 1:1000 dilution, Cat No. A-11008, Invitrogen) was applied for 1 h. After being washed twice with PBS, cells were stained with DAPI (Sigma, USA). All images were acquired on Nikon A1Si Laser Scanning confocal Microscope (Nikon Instruments Inc., Japan).

## Lentivirus, siRNA, plasmid construction, and cell transfection

Lentiviral vector containing human full-length circCDK13, the mutant version of circCDK13 (the sequence of "ACAAACA" was replaced with "UGUUUGU"), as well as their corresponding negative control were purchased from Geneseed (Guangzhou, China). Lentiviral vector containing IGF2BP3 and their corresponding negative control were purchased from GenePharma (Shanghai, China). si-circCDK13, si-IGFB2P3, and si-METTL3/14 were purchased from GenePharma (Shanghai, China). For the transfection of siRNAs, cells were transfected using the

Lipofectamine 3000 kit (Invitrogen, Carlsbad, CA, USA) according to the manufacturer's instructions. All sequences are listed in Supplementary Table 3.

## RNA pull-down assay

Biotinylated circCDK13 and its anti-sense sequence were synthesized by GenePharma (Shanghai, China). RNA pull-down assays were performed using a PureBinding® RNA-Protein pull-down Kit (Geneseed, Guangzhou, China) according to the manufacturer's instructions. Approximately $1 \times 10^7$ cells were collected, disrupted, and incubated with 100 µl of streptavidin-coated magnetic beads for 30 min at 4 °C, with rotating at 10 rpm/min, with biotin-labeled circCDK13 probe. The cell lysates were incubated with streptavidin-coated magnetic beads to pull down the biotin-labeled RNA complex. The eluted proteins were separated by SDS-PAGE followed by silver staining (Fast Silver Stain Kit, Beyotime, China). Afterwards, the silver-stained differential protein band in gel was cut off and subjected to digestion with trypsin at 37 °C overnight. The enzyme-digested polypeptide samples were dried and dissolved again for liquid chromatography-tandem mass spectrometry (LC-MS/MS) analysis (MaxQuant 1.6.17.0.). The specific sequences of the probes can be found in Supplementary Table 4.

## RNA immunoprecipitation (RIP)

PureBinding®RNA Immunoprecipitation Kit (Geneseed, Guangzhou, China) was used to perform RIP experiments according to the manufacturer's instructions. HDFs and HEKs ($1 \times 10^7$) were lysed in 1 ml of RIP lysis buffer with RNase inhibitors. The cell lysates were then incubated with beads coated with IgG, anti-IGFB2P3 on a rotator at 4 °C overnight. The coprecipitated RNAs were examined using RT-qPCR.

## Western blot assay

Cells and tissues were collected and lysed with RIPA buffer containing proteinase inhibitor. The protein concentration was measured with BCA Protein Quantification Kit (Solarbio, China). Proteins were separated on SDS-PAGE and electro-transferred to nitrocellulose membranes (Millipore). After blockage, the membranes were incubated at 4 °C overnight with primary antibodies including IGF2BP3 (1:2000 dilution, Cat No.14642-1-AP, ProteinTech), c-MYC (1:5000 dilution, Cat No. 67447-1-Ig, ProteinTech), CD44 (1:2000 dilution, Cat No. 60224-1-Ig, ProteinTech), Cyclin D1 (1:5000 dilution, Cat No. 60186-1-Ig, ProteinTech), CDK13 (1:1000 dilution, Cat No. A10258, ABclonal), GAPDH (1:100,000 dilution, Cat No. 60004-1-Ig, ProteinTech), β-actin (1:1000 dilution, Cat No. ab8227, Abcam). Peroxidase-conjugated (HRP)-linked secondary antibody (Goat Anti-Rabbit IgG, 1:10,000 dilution, Cat No. HS101-01, Goat Anti-Mouse IgG, 1:10,000 dilution, Cat No. HS201-01, Transgen, China) was used to incubate with these membranes for 1 h at room temperature. The antigen–antibody reaction was visualized via an ECL kit (Thermo Fisher Scientific, USA) and imaged by UVITEC Alliance micro Q9 system (UVITEC, Britain).

## Gene-specific m⁶A RT-qPCR

The $m^6A$ modifications on circCDK13 were determined using the BersinBioTM MeRIP Kit (BersinBio, Guangzhou, China) according to the manufacturer's instructions. Briefly, 100 µg of total RNAs were fragmented to approximately 300 nt in length by metal-ion-induced fragmentation. Next, RNA was incubated with 5 µg anti-m6A antibody (1:1000 dilution, Cat No. 68055-1-Ig, Proteintech) or IgG for 2 h at 4 °C. Protein A/G magnetic beads were mixed with the antibody-treated RNA in IP buffer for 2 h at 4 °C. Then, the bound RNAs were washed and eluted with Proteinase K and elution buffer for 1 h at 55 °C. Finally, RNA bound to immunoprecipitated proteins was extracted with Phenol-Chloroform-isoamyl alcohol. RT-qPCR was performed to measure the methylated RNA expression levels. The related enrichment of m⁶A in each sample was calculated by normalizing to input.

## Transfection and preparation of sEVs

CP-MSCs were seeded in T25 culture flask and incubated at 37 °C in a humidified atmosphere with 5% $CO_2$. When the cells reached 60%−70% confluence, they were co-transfected with the vector or circCDK13 lentivirus (Geneseed, Guangzhou, China) according to the manufacturer's instructions. 48 h after transfection, puromycin (2 µg/ml, Beyotime, China) was added to the cell culture medium to obtain stable strains. sEVs were prepared from the supernatant fluids of Vector-CP-MSCs or circCDK13-CP-MSCs by differential centrifugation (Supplementary Fig. 16a). Briefly, the supernatant underwent a series of centrifugation steps at varying speeds and durations, namely 300 g for 10 min, 3000 g for 15 min, and 10,000 g for 60 min at 4 °C, to remove cells and debris, and then filtered using a 0.22 µm filter (Millipore, USA). The filtrate was then subjected to ultracentrifugation at 200,000 g for 90 min at 4 °C using an XPN-100 ultracentrifuge (Beckman Coulter, USA). Subsequently, the pellet was resuspended in PBS and then ultracentrifuged again at 200,000 g for 90 min. Ultimately, the resulting pellet was resuspended in PBS for further study.

## Characterization of sEVs

Well-established markers of purified sEVs were verified by western blot analysis. The following antibodies were used: anti-CD63 (1:1000 dilution, Cat No. 25682-1-AP, Proteintech), CD9 (1:2000 dilution, Cat No. 20597-1-AP, Proteintech), TSG101 (1:10,000 dilution, Cat No. 28283-1-AP, Proteintech), Alix (1:20,000 dilution, Cat No. 12422-1-AP, Proteintech), Calnexin (1:10,000 dilution, Cat No. 66903-1-Ig, Proteintech). The morphology of sEVs was examined by transmission electron microscope (TEM, H-7650C, Hitachi, Japan). Briefly, the purified sEVs were fixed in 4% paraformaldehyde (PFA), 10 µl of sEVs solution was dropped on the copper grid, incubated at room temperature for 10 min. After washing three times with sterile distilled water, 10 µl of 2% uranyl acetate was dropped on the copper grid for negative staining for 1 min, the floating liquid was absorbed by filter paper, and dried under an incandescent lamp for 2 min. Finally, the particle morphology was visualized by TEM at 80 kV. The size distribution and concentration of sEVs were measured using Nanoparticle tracking analyzer (NTA, Particle Metrix, Germany).

## Absolute quantitative polymerase chain reaction (qPCR)

Absolute qPCR was employed to determine the copy number of circCDK13 in both overexpressed circCDK13 CP-MSCs and sEVs. To accomplish this, a 220 bp characteristic segment containing the back-spliced junction of circCDK13 transcript was cloned and inserted into the pUC57 vector (HITRO BioTech, Beijing). Various concentrations of plasmid were utilized to construct a standard curve through real-time PCR. The copy number of standard compounds was determined using the following formula: copy number (copy/µl) = $6.02 \times 10^{23}$ × plasmid concentration (ng/µl) × $10^{-9}$/ [(molecular weight of vector + molecular weight of inserted fragment) × 660] (g/mol). The amplification curve and standard curve equation were derived, with the equation represented as $Y = aX + b$ [log value on 10 of the initial amplification copy number was the abscissa (X), and the corresponding cycle number was the ordinate (Y)]. Cycle threshold (Ct) values of circCDK13 were calculated to arbitrary units using a linear equation based on the plasmid standard allowing conclusions for the copy number of the circCDK13.

## RNA protection assay

circCDK13$^{OE}$-sEVs ($1.02 \times 10^{12}$ particles) were incubated with a mixture of 20 µl RNase A/T1 (EN0551, Thermo, USA) with or without 1% Triton X-100 at 37 °C for 30 min. Following inactivation at 75 °C for 5 min, RNA was extracted using the Steady Pure Rapid RNA Isolation Kit (Accurate Biology, China). Finally, the absolute qPCR analysis was employed to estimate the copy number of circCDK13.

## Uptake of sEVs

CM-DiI (C7001, Invitrogen, USA) was utilized to label sEVs following the manufacturer's instructions. Briefly, sEVs were co-incubated with CM-DiI at room temperature for 30 min, followed by centrifugation at 200,000 g, 4 °C for 90 min to remove unbound CM-DiI dye, resuspended the pellet in PBS and centrifuged again. Subsequently, the pellet containing CM-DiI-labeled sEVs was co-incubated with HDFs (or HEKs) at 37 °C for 6 h, then the cells were washed with PBS and fixed with 4% paraformaldehyde. The internalization of sEVs by HDFs (or HEKs) was assessed using Phalloidin-Rhodamine B for cytoskeleton staining and DAPI for cell nucleus staining. The observations were made using a Nikon A1Si Laser Scanning confocal microscope (Nikon Instruments Inc., Japan).

## In Vivo administration of sEVs

Male diabetic mice (BKS-Leprem2Cd479/Gpt, db/db, 10−12 weeks of age) purchased from Jiangsu Jicui Yaokang Biotechnology Co., Ltd. (NanJing, China), male SD rats (6−8 weeks of age) purchased from SiPeiFu biotechnology Co., Ltd (Beijing, China), and all rodents were housed in a suitable environment with 25 °C, 50%−45% humidity, and 12 h dark/light cycle, given free access to water and food. Experimental procedures were approved by the Institutional Animal Care and Use Committee of Chinese PLA General Hospital and performed in accordance with the Animal Research: Reporting of In Vivo Experiments (ARRIVE) guidelines. The sample size required for animal research was based on previous experimental results and was similar to the sample size commonly used in the field. The mice and rats were coded and randomly divided into experimental and control groups.

Type I diabetic rats were induced by intraperitoneal injection of streptozotocin (STZ) into healthy rats. After 12 h of fasting subjects, SD rats were induced by intraperitoneal injection of STZ (65 mg/kg; Sigma, Missouri, USA) dissolved in pH 4.5 citrate buffer. Then blood glucose was measured every 7 d. The rats with blood glucose concentration greater than 16.7 mM after 4 weeks were determined as diabetic rats. Animals were anesthetized with Pentobarbital sodium, hairs were clipped, skin site was disinfected and a pair of full thickness excisional wounds (diameter = 1.0 cm) were created on the dorsal region of each rodent using a punch. Afterward, wounds were randomly assigned into Control groups, N-sEVs groups, and circCDK13$^{OE}$-sEVs groups ($n = 6$). PBS (100 μl), N-sEVs (100 μl, $2 \times 10^{11}$ particles/ml), and circCDK13$^{OE}$-sEVs (100 μl, $2 \times 10^{11}$ particles/ml) were subcutaneously injected around the wounds at 4 sites (25 μl per site) on day 0, 3, 7, 10, 14, and 17 post-wounding. The wounds were photographed and analyzed by Image J software on days 0, 3, 7, 10, 14, and 21 post-wounding. The animals were euthanized with an overdose of pentobarbital sodium on days 3, 7, 14, and 21, and wound samples from adjacent skins were collected, fixed in 4% paraformaldehyde for histological examination or quick-frozen in liquid nitrogen and stored at −80 °C for further analysis.

## Histopathological analysis of wounds

The obtained wound tissues were fixed with 4% paraformaldehyde for at least 24 h, then dehydrated through a graded series of ethanol and embedded in paraffin. The embedded tissues were sectioned to 4 μm thick sections, subsequently, hematoxylin and eosin (H&E) and Masson's trichrome staining were conducted according to the manufacturer's instructions. Images were captured using a fully automatic digital pathology scanner (UNIC PRECICE 610, Suzhou, China). Wound width and epithelium length were measured using Image J software (Version 1.53k).

## Tissue immunofluorescence Staining

Skin tissue samples were fixed with 4% paraformaldehyde, embedded in paraffin, and sectioned. The 4 μm thick sections for antigen retrieval were permeabilized for 20 min with 0.5% Triton X-100. Subsequently, the slides were incubated with 5% BSA for 60 min at RT. The slides were stained by incubation with primary antibodies including IGF2BP3 (1:100 dilution, Cat No. 14642-1-AP, ProteinTech), c-MYC (1:1000 dilution, Cat No. 67447-1-Ig, ProteinTech), CD44(1:200 dilution, Cat No. 60224-1-Ig, ProteinTech), Vimentin (1:1000 dilution, Cat No. 60330-1-Ig, ProteinTech) at 4 °C overnight. Then, the slides incubated with Goat anti-Rabbit IgG (H + L) Cross-Adsorbed Secondary Antibody, Alexa Fluor™ 488 (1:1000 dilution, Cat No. A-11008, Invitrogen) and Goat anti-Mouse IgG (H + L) Highly Cross-Adsorbed Secondary Antibody, Alexa Fluor™ 594 (1:1000 dilution, Cat No. A-11032, Invitrogen) for 60 min at RT. Finally, the slides were stained with DAPI to visualize the nuclei. All images were acquired on Nikon A1Si Laser Scanning confocal Microscope (Nikon Instruments Inc., Japan).

## Statistical analysis

Statistics as well as graphical representations were performed using GraphPad Prism™ 8.0 software (GraphPad Software Inc., USA). All the results in this study were shown as the means ± standard deviation (SD). At least three independent experiments were performed for all experiments and representative images are shown. Comparisons between two groups were performed using Student's $t$ test. Comparisons between more than two groups were analyzed by a one-way ANOVA test. Results were considered statistically significant when $P < 0.05$.

## Reporting summary

Further information on research design is available in the Nature Portfolio Reporting Summary linked to this article.

## Data availability

Source data are provided with this paper. The remaining data are available from this paper and supplementary materials or the corresponding authors upon request. The circRNA sequencing data obtained from wounds are publicly available on the GEO database under accession code (GSE114248). Source data are provided with this paper.

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

## Acknowledgements

This study was supported by the Natural Science Foundation of Beijing (7242129), the National Nature Science Foundation of China (82172211, 92268206, 82172231), the National Key Research and Development Programs of China (2022YFA1104300), the CAMS Innovation Fund for Medical Sciences (CIFMS, 2019-I2M-5-059), the Military Medical Research Projects (2022-JCJQ-ZB-09600; 2023-JSKY-SSQG-006), and the 1·3·5 Project for Disciplines of Excellence, West China Hospital, Sichuan University (ZYGD22008).

## Author contributions

Q.H., Z.C. and Z.W. contributed equally to this study. Q.H. and C.Z. conceived and designed the experiments. Q.H., Z.C. and Z.W. performed most of the experiments. S.M. and S.C. performed part of animal experiments. Q.W., Q.L., Y.L., K.M., W.H., W.Z. and Y.Q. provided technical and material support. Q.H. analyzed data and drafted the manuscript. H.L., X.F. and C.Z. initiated the study and reviewed the manuscript. All authors have read and approved the article.

## Competing interests

The authors declare no competing interests.
