## [Peer Review File · Nature Communications]

REVIEWER COMMENTS

Reviewer #1 (Remarks to the Author):

In the present study, Huang et al. proposed a novel treatment strategy for diabetic wound by investigating the possible role of certain circular RNAs in the pathogenesis and treatment of wound healing in diabetic conditions. The authors presented data showing that circCDK13 was downregulated in diabetic wounds. Depletion of circCDK13 in human dermal fibroblasts and human epidermal keratinocytes inhibited the capacities of their proliferation and migration. In animal models, engineered small extracellular vesicles with overexpression of circCDK13 could accelerate the wound healing process in diabetic animals. The idea was relative novel and the research was with good experimental design. However, the study heavily relied on overexpression of the circular RNA embedded in the extracellular vesicles. It is not clear for the physiological function of this circular RNA in wound repair. Here are some additional comments.

1. For rationale, the authors analyzed the microarray data and found 111 circRNAs including circCDK13 were downregulated. However, it was not explained why circCDK13 was chosen for further investigation. Is circCDK13 the most decreased circRNA among these 111?

2. For FigS3 which showing the cell migration, the yellow lines indicating the migrating distance are misleading, especially for the 24 h time point, which exaggerated the changes between groups. The yellow lines should drawn on the line of most cells migrated, but not on the line of the several cells that migrated fastest. There should be a same standard for all the groups.

3. The labeling of Fig1 and Fig2 is totally messed up. The panels of Fig 1 do not match the description in the Results and Figure Legends. "The expression of circCDK13 was down-regulated in HDFs and HEKs treated with AGE-BSA" cannot be found as stated in Fig 1j, k. The Sanger sequencing of back splice junction site of circCDK13 cannot be found in Fig 1l. There are more similar mistakes likes this. The authors should double check whether they upload a wrong figure file.

4. The statement in "line 170-175" about the translation is not accurate. The authors drew the conclusion that "the observed promotion of proliferation and migration mediated by circCDK13 is not attributed to the encoded protein CDK13" based on the results that CDK13 protein (at 164 kD) levels were not changed by overexpression or knockdown of circCDK13. However, the authors should know that the MW of circRNA encoded proteins should be smaller than the full-length mRNA encoded proteins. If the author would like to claim that the cell function changes were not mediated by circCDK13 encoded protein, the author should not examine the CDK13 protein at 164kD but should predict the

possible MW of circCDK13 encoded protein and examine the proteins at the predicted MW using CDK13 antibodies.

5. The concentration and duration of cycloheximide treatment were not clear stated in the Methods or Figure legends section.

6. It is stated in line 271 that “circCDK13 and IGF2BP3 synergistically enhanced the stability of CD44 and c-MYC mRNA”. I wonder whether the authors examined whether circCDK13 or IGF2BP3 directly bind with CD44 and c-MYC mRNA. If not, how circCDK13 and IGF2BP3 affected the stability of CD44 and c-MYC mRNA? Is there any mediator involved in?

Reviewer #2 (Remarks to the Author):

This study aims to identify a novel circular RNA (circRNA) loaded into small extracellular vesicles (sEVs) to reverse the impaired wound healing caused by diabetes.

Utilizing an external database, the investigators identified circCDK13 that was downregulated in human diabetic wounds. In support of circCDK13, they show depletion of circCDK13 reduces the migration and proliferation of human dermal fibroblasts (HDF) and human epidermal keratinocytes, and overexpression showed the opposite. At a mechanistic level, circCDK13 interacted directly with IGF2BP3 in an m⁶-methyladenosine manner to increase the expression of CD44 and c-Myc. Encapsulation of mesenchymal stem cell-derived small extracellular vesicles (sEVs) that overexpress circCDK13 (circCDK13OE-sEVs) reversed the effects of experimental hyperglycemia (Age-BSA) on the migration and proliferation of HDFs and HEKs. circCDK13OE-sEVs were more effective than N-sEVs in accelerating wound healing and skin appendage regeneration in db/db type 2 diabetic rats and Streptozotocin (STZ)-induced type I diabetic rats.

The experiments showing how circCDK13 interacts with IGF2BP3 and their effects on HDFs and HEKs under hyperglycemic conditions were elegant and convincing. While the circCDK13OE-sEVs appeared to increase the wound healing rate, the design and interpretation raised concerns. Given that the primary goal of these experiments was the development of a therapeutic to reverse wound healing in diabetics, these experiments need to be more detailed.

1. It is stated that circCDK13 was downregulated in “diabetic wounds compared with acute wounds”. Do you mean non-diabetic wounds, or are you referring to acute or chronic diabetic wounds?
2. The optimal experimental wound is stented because rats have a subcutaneous muscle, the panniculus carnosum, that causes the wound to heal by contraction. Stenting the wound promotes a healing pattern more typical of that seen in humans
3. None of the wounding experiments were performed to complete healing, which is the proper standard. This is important for several reasons. The primary goal of these experiments is to evaluate a therapeutic intervention. The acceleration of the wound healing rate measured in days is the benchmark to determine if an intervention justifies its cost. None of the experiments contained a wild-type rat control group.
4. Wound healing progresses between multiple defined stages: proliferative, angiogenic and remodeling. Understanding the role of circCDK13 during these stages would greatly enhance the significance of the experiments.
5. The wound healing rate in the streptozocin-induced type 1 diabetes was only carried out for 14 days. Why? According to the protocol, the intervention was administered until day 17. Why exclude that dose? Moreover, the differences in healing rates were very similar in the N-sEVs and the circCDK13OE-sEVs suggesting a minimal therapeutic effect.
6. Given the in vivo model, why not measure the expression of circCDK13 during the different stages of wound healing in diabetic and wild-type rats? Or use a siRNA against circCDK13 in wild-type rats?

Reviewer #3 (Remarks to the Author):

In the present study, author revealed that impaired healing of diabetic wounds was tightly associated with the downregulation of circCDK13, and then confirmed a novel wound healing-promoting mechanism that circCDK13 directly interacted with IGF2BP3 to form a circRNA-protein-mRNA ternary complex, which synergistically enhanced the stability of CD44 and c-MYC mRNA to promote the proliferation and migration of HDFs and HEKs. Author successfully constructed engineered sEVs bearing circCDK13 and corroborated that circCDK13OE-sEVs. Author confirmed that in the wounds of db/db diabetic mice and STZ-induced type I diabetic rats, circCDK13OE-sEVs could accelerate re-epithelialization and granulation tissue formation, and promote wound remodeling and regeneration of skin appendages.

1. The co-localization experiments of circCDK13 and IGF2BP3 using Fish showed that circCDK13 and IGF2BP3 interact. Please provide more data to support this finding.

2. The authors perform immunoprecipitation of circCDK13 followed by mass-spectrometry to identify protein interactors. These data are not shown: Fig. S7 just shows the LC-MS/MS plot, while a supplementary table with the identified interactors should be provided.
3. Authors should examine studies of circCDK13OE-sEVs on function in other organs (e.g., kidneys, etc.)
4. The authors should examine the effect of overexpression or knockdown of circCDK13 on cell death (such as autophagy, apoptosis, necrosis, etc.) in HDFs and HEKs.
5. Other studies on circCDK13 have shown that activation of endogenous CDK13 can promote the expression of circCDK13, and the high expression of circCDK13 promotes the occurrence and development of prostate cancer, while inhibiting the expression of circCDK13 can reduce the occurrence of prostate cancer. Authors should consider whether the expression of circCDK13 might produce tumorigenicity.◦
6. The authors should increase the detection of whether circCDK13 affects classical proliferation pathways (such as Yap).
7. Macrophages play an indispensable role in the process of wound repair in the body. The authors should examine the expression of circCDK13 and IGF2BP3 in macrophages and its influence on them.
8. Most chronic wounds are characterized by a large number of inflammatory cell infiltration, leading to overexpression of reactive oxygen species and MMP. It is suggested that the authors examine the effects of overexpression or knockdown of circCDK13 on the expression of reactive oxygen species and MMP.
9. Please revise the English grammar and writing style of the manuscript, spelling and grammatical errors should be excluded.

Point-by-point responses to the comments from Reviewers

We sincerely thank all the Reviewers for their constructive comments and helpful suggestions. We have done our best to address all the issues raised and hope that our revised manuscript now meets your expectations

Reviewer #1:

In the present study, Huang et al. proposed a novel treatment strategy for diabetic wound by investigating the possible role of certain circular RNAs in the pathogenesis and treatment of wound healing in diabetic conditions. The authors presented data showing that circCDK13 was downregulated in diabetic wounds. Depletion of circCDK13 in human dermal fibroblasts and human epidermal keratinocytes inhibited the capacities of their proliferation and migration. In animal models, engineered small extracellular vesicles with overexpression of circCDK13 could accelerate the wound healing process in diabetic animals. The idea was relative novel and the research was with good experimental design. However, the study heavily relied on overexpression of the circular RNA embedded in the extracellular vesicles. It is not clear for the physiological function of this circular RNA in wound repair. Here are some additional comments.

Response: Thanks very much for your high evaluation.

1. For rationale, the authors analyzed the microarray data and found 111 circRNAs including circCDK13 were downregulated. However, it was not

explained why circCDK13 was chosen for further investigation. Is circCDK13 the most decreased circRNA among these 111?

Response: Thanks very much for reviewer's question. The reasons for choosing circCDK13 for in-depth study in this study are as follows. Firstly, we analyzed a set of microarray data (GSE114248) published by Wang[1], and found that 111 circRNAs including circCDK13 were down-regulated in the diabetic foot ulcer (DFU) compared with normal wounds (NWs). Secondly, we carried out homology analysis on the top 40 circRNAs with the highest expression differences and found that 6 circRNAs shared homology between Homo sapiens and mice. Thirdly, we used AGE-BSA, an alternative to AGEs commonly used to simulate the diabetic wound microenvironment, to intervene HEKs and HDFs and found that three circRNAs, including circCDK13, were significantly downregulated in the AGE-BSA intervention group. Subsequently, we used siRNA interference technology to knock down the intracellular expression abundance of above three circRNAs and observed that the proliferation and migration of HEKs and HDFs were significantly reduced only by knocking down circCDK13. In addition, circCDK13 was also reported to play a significant role in the proliferation, invasion, and metastasis of cancer cells[2]. So, in our study, circCDK13 was chosen for further investigation. The results were described on Page 5 Line 114-Page 6 Line 139 and screening process of circCDK13 was shown in Supplementary Fig.1c.

Page 5 Line 114- Page 6 Line 139:

CircCDK13 was chosen as a promising therapeutic agent for diabetic wounds

CircRNAs have been discovered as powerful actors of gene expression with critical

functions in many diseases. To elucidate the role of circRNAs in the healing process of diabetic wounds, we analyzed a set of microarray data (GSE114248) published by Wang⁸, and found that 111 circRNAs including circCDK13 were down-regulated in the DFU compared with normal wounds (NWs) (Supplementary Fig. 1a). Heat map of the top 40 circRNAs was shown in Fig. 1a and normalized expression level of circCDK13 were calculated with the microarray data (Fig. 1b). Homology analysis was performed on the top 40 circRNAs and we found that 6 circRNAs including circCDK13 shared homology between Homo sapiens and mice. The homologous sequence of circCDK13 was shown in Supplementary Fig. 1b. Additionally, the lower expression of circCDK13 was also observed in the wounds of diabetic mice (Fig. 1c). To simulate the microenvironment of diabetic wounds *in vitro*, AGE-BSA, an alternative to AGEs commonly used in research, was produced by co-incubating D-glucose with bovine serum albumin (BSA) at 37°C for 2 months (Supplementary Fig. 2a-c). After treated with AGE-BSA at different concentrations, HDFs and HEKs exhibited the decreased proliferation assessed by CCK8 assay (Fig. 1d, e) and migration assessed by scratching test (HDFs: Supplementary Fig. 3a, Fig. 1f; HEKs: Supplementary Fig. 4a, Fig. 1g) and transwell assay (HDFs: Supplementary Fig. 3b, Fig. 1h; HEKs: Supplementary Fig. 4b, Fig. 1i). Among above 6 circRNAs, we found that three circRNAs, including circCDK13, were significantly downregulated in AGE-BSA treated HDFs and HEKs. Fig. 1j, k demonstrated the downregulation of circCDK13 in AGE-BSA-treated HDFs and HEKs. Among the three downregulated circRNAs, circCDK13 was reported to play a significant role in the regulation of cellular proliferation and migration¹⁸. Therefore,

we speculated that the downregulation of circCDK13 might be involved in the mechanisms of impaired wound healing in diabetes, implying circCDK13 was a promising therapeutic agent for diabetic wounds. The screening process of circCDK13 was summarized in Supplementary Fig. 1c.

Supplementary Fig. 1c:

Supplementary Fig. S1. (c) Schematic diagram of the circCDK13 screening process.

References

- [1] Wang A, Toma MA, Ma J, et al. Circular RNA hsa_circ_0084443 Is Upregulated in Diabetic Foot Ulcer and Modulates Keratinocyte Migration and Proliferation. *Adv Wound Care (New Rochelle)*. 2020;9(4):145-160.
- [2] Qi JC, Yang Z, Lin T, et al. CDK13 upregulation-induced formation of the positive feedback loop among circCDK13, miR-212-5p/miR-449a and E2F5 contributes to prostate carcinogenesis. *J Exp Clin Cancer Res*. 2021;40(1):2.

2. For FigS3 which showing the cell migration, the yellow lines indicating the migrating distance are misleading, especially for the 24 h time point, which exaggerated the changes between groups. The yellow lines should drawn on the line of most cells migrated, but not on the line of the several cells that migrated fastest. There should be a same standard for all the groups.

Response: Thank you for your insightful review and valuable suggestion. We relabeled the yellow lines indicating the migrating distance in Supplementary Fig. S3a.

Supplementary Fig. S3a:

Supplementary Fig. S3. (a) Representative images of HDFs migration measured by wound healing assay at 0 h, 12 h, and 24 h. scale bar, 400 μ m.

3. The labeling of Fig1 and Fig2 is totally messed up. The panels of Fig 1 do not match the description in the Results and Figure Legends. “The expression of circCDK13 was down-regulated in HDFs and HEKs treated with AGE-BSA” cannot be found as stated in Fig 1j, k. The Sanger sequencing of back splice junction site of circCDK13 cannot be found in Fig 1l. There are more similar mistakes likes this. The authors should double check whether they upload a wrong

figure file.

Response: Thank you for your careful review. In accordance with your request, we have double checked the labeling of Fig 1 and Fig 2 and the matching between panels of Fig 1 and the description in the Results and Figure Legends. As shown in Fig. 1j, k, the expression abundance of circCDK13 in HDFs and HEKs gradually decreased as the concentration of AGE-BSA increased. “The expression of circCDK13 was down-regulated in HDFs and HEKs treated with AGE-BSA” means the comparison between AGE-BSA group and control group (AGE-BSA concentration of 0 $\mu\text{g/ml}$). To avoid misunderstanding, AGE-BSA concentration of 0 $\mu\text{g/ml}$ group has been changed to control group in Fig. 1j, k. Sanger sequencing following PCR was used to show the “head-to-tail” splicing of circCDK13 and the Sanger sequencing of back splice junction site of circCDK13 can be found in Fig. 1l, lower right panel. To match the description in the results and Fig. 1n-q, we changed CDK13 to CDK13 mRNA in Fig. 1n-q. Additionally, to better understand the description of the results, we reordered Fig. 1p-t as following and rewrote the description of Fig. 1a-k on Page 5 Line 114-Page 6 Line 139 (See the Response to the review’s first question) and Fig. 1n-t on Page 7 Line 149-154. To match panels of Fig. 1, Figure Legend of Fig. 1 was rewritten. As your required, Fig. 2 was reordered and Figure Legend was rewritten. Some results were separated from Fig. 2 to compose Supplementary Fig. 5. Additionally, the corresponding description of Fig. 2 and Supplementary Fig. 5 was also rewritten on Page 7 Line 158-Page 8 Line 174.

Page 7 Line 149-154:

Actinomycin D assay (Fig. 1n, o) and RNase R exonuclease digestion assay (Fig. 1p, q) further indicated that circCDK13 was more stable than CDK13 mRNA. Furthermore, fluorescence in situ hybridization (FISH) (Fig. 1r) and nuclear and cytoplasmic fractionation analysis (Fig. 1s, t) demonstrated that circCDK13 was predominantly localized in the cytoplasm. Collectively, these results implicated that circCDK13 in HDFs and HEKs is a stable circRNA generated from CDK13 by back-splicing.

Page 7 Line 158- Page 8 Line 174:

Two siRNAs (Fig. 2a) targeting the back-splicing sequence of circCDK13 were used to knockdown the expression of circCDK13 in HDFs (Fig. 2b) and HEKs (Fig. 2c). CCK-8 assay was performed to show that the knockdown of circCDK13 resulted in the decreased viability of HDFs (Fig. 2d) and HEKs (Fig. 2e). Next, an overexpression vector of circCDK13 was constructed (Supplementary Fig. 5a, b) and then transfected into HDFs and HEKs (Fig. 2f) to overexpress circCDK13 (Fig. 2g, h). The results of CCK-8 assay show that the enforced expression of circCDK13 dramatically enhanced the viability of HDFs (Fig. 2i) and HEKs (Fig. 2j). Furthermore, the percentage of EdU-positive cells was decreased by circCDK13 knockdown and increased by circCDK13 overexpression in HDFs (Fig. 2k, Supplementary Fig. 5c) and HEKs (Fig. 2l, Supplementary Fig. 5d). Subsequently, scratch and transwell assays were performed to investigate the effect of circCDK13 on cell migration. The migration area of circCDK13 knockdown group was remarkably lower than that of its control group and the migration area of circCDK13 overexpression group was higher than that of corresponding control group (Fig. 2m, n; Supplementary Fig. 5e, f). Additionally, the

results of transwell assay demonstrated that circCDK13 knockdown decreased and circCDK13 overexpression increased the number of migrated cells (Fig. 2o, p; Supplementary Fig. 5g, h).

Fig. 1 and Fig.1 legend:

Fig.1 CircCDK13 was downregulated in diabetic wounds. **a** Heat map of the top 40 circRNAs with significant differential expression. Red represents up-regulated circRNAs, blue represents down-regulated circRNAs in diabetic foot ulcer (DFU) vs normal wounds (NWs). **b** The normalized expression levels of circCDK13 were extracted from the microarray data. **c** The results of RT-qPCR analysis showed that the

expression of circCDK13 was down-regulated in the wound of diabetic mice (n=8). DWs, diabetic mice wounds; WWs, Normal wild-type mice wounds. **d, e** CCK-8 assay showed the proliferative ability of HDFs and HEKs treated with different concentrations of AGE-BSA (n=6). **f, g** Quantification of the mobility rate of HDFs and HEKs in wound healing assay (n=5). **h, i** The number of migrated cells in transwell migration assay (n=5). **j, k** Relative expression levels of circCDK13 in HDFs and HEKs treated with different concentrations of AGE-BSA (n=3). **l** Schematic illustration showing the circularization of CDK13 exon 2 forming circCDK13. Sanger sequencing following PCR was used to show the “head-to-tail” splicing of circCDK13. **m** The presence of circCDK13 was validated in HDFs and HEKs by RT-qPCR. Divergent primers amplified circCDK13 in cDNA but not in genomic DNA. GAPDH was used as a negative control. **n, o** Actinomycin D treatment was used to evaluate the stability of circCDK13 and CDK13 mRNA in HDFs and HEKs. **p, q** Resistance of circCDK13 and CDK13 mRNA to digestion with RNase R exonuclease was detected by RT-qPCR (n=3). RNase R exonuclease specifically degraded linear RNAs but not circRNAs. **r** Localization of circCDK13 detected by fluorescence in situ hybridization (FISH) assay in HDFs and HEKs. Nuclei were stained with DAPI. Scale bar = 50 μ m. **s, t** Nuclear and cytoplasmic fractionation revealed the level of circCDK13 in the nucleus and cytoplasm of HDFs and HEKs (n=3). U6 and GAPDH were used as positive controls in the nucleus and cytoplasm, respectively (n=3). Data are represented as mean \pm SD. Statistical significance is indicated as follows: NS, not significant, *p < 0.05, **p < 0.01, ***p < 0.001.

Fig.2 and Fig.2 legend:

Fig.2 CircCDK13 promotes the proliferation and migration of HDFs and HEKs

in vitro. **a** Schematic representation and target sequences of the siRNAs specific to the back splice junction of circCDK13. **b, c** RT-qPCR results showed that two siRNAs targeting the back-splicing site of circCDK13 significantly decreased the expression levels of circCDK13 in HDFs and HEKs (n=3). **d, e** CCK-8 assay showed that knockdown of circCDK13 induced the repression of viability of HDFs and HEKs (n=6). **f** HDFs and HEKs expressed green fluorescent protein (GFP), indicating that pLC5-ciR plasma was successfully introduced into cells and expressed stably. **g, h** RT-

qPCR was performed to detect circCDK13 after overexpression of circCDK13 in HDFs and HEKs (n=3). **i, j** CCK-8 assay showed that overexpression of circCDK13 upregulated viability of HDFs and HEKs (n=6). **k, l** EdU incorporation assay was performed to assess DNA synthesis in HDFs and HEKs. Red fluorescence represented EdU-positive cells, while blue fluorescence represented total cells. Scale bar, 100 μ m. **m, n** Wound healing assay was performed to measure the motility of HDFs and HEKs. Scale bar, 200 μ m. **o, p** Representative images of transwell migration assay of HDFs and HEKs. Scale bar, 50 μ m. Data are represented as mean \pm SD. Statistical significance is indicated as follows: NS, not significant, * $p < 0.05$, ** $p < 0.01$, *** $p < 0.001$.

Supplementary Fig. 5 and Fig. 5 legend:

Supplementary Figure 5. CircCDK13 promotes the proliferation and migration of HDFs and HEKs. (a) Schematic diagram of the plasmid structure used to construct the circCDK13 overexpression system. The pLC5-ciR plasmid was provided by Genesee

Biotechnology Co., Ltd (Guangzhou, China). **(b)** Structures of the circCDK13 expression construct and the control vector. **(c, d)** Quantification histogram represented EdU positive cell percentage (n=5). **(e, f)** Quantification histogram represented migration rate (n=5). **(g, h)** The number of migrated cells in transwell migration assay (n=5). Data are represented as mean \pm SD. Statistical significance is indicated as follows: ***p < 0.001.

4. The statement in “line 170-175” about the translation is not accurate. The authors drew the conclusion that “the observed promotion of proliferation and migration mediated by circCDK13 is not attributed to the encoded protein CDK13” based on the results that CDK13 protein (at 164 kD) levels were not changed by overexpression or knockdown of circCDK13. However, the authors should know that the MW of circRNA encoded proteins should be smaller than the full-length mRNA encoded proteins. If the author would like to claim that the cell function changes were not mediated by circCDK13 encoded protein, the author should not examine the CDK13 protein at 164kD but should predict the possible MW of circCDK13 encoded protein and examine the proteins at the predicted MW using CDK13 antibodies.

Response: Thank you for your careful review. We strongly agree with your opinion.

The conclusion that “the observed promotion of proliferation and migration mediated by circCDK13 is not attributed to the encoded protein CDK13” based on the results that CDK13 protein (at 164 kD) levels were not changed by overexpression or knockdown

of circCDK13 is not accurate. As you said, if circCDK13 can encode protein CDK13, the MW of circRNA encoded proteins should be smaller than the full-length mRNA encoded proteins. However, in literature, to date, there have been no reports of circCDK13-encoded protein, which needs an in-depth study. In our study, we detected the CDK13 mRNA and protein expressions (Supplementary Fig. 6) and found that they were unaffected by siRNAs and overexpression vector of circCDK13. The results demonstrated that siRNAs used in our study didn't degrade CDK13 linear mRNA transcript and circCDK13 couldn't increase expression of CDK13 mRNA and protein by a positive feedback pathway reported before[1]. Collectively, the above findings indicated that circCDK13 rather than CDK13 protein played an essential role in regulating the proliferation and migration of HDFs and HEKs. The results were described on Page 8 Line 174- Page 8 Line 181.

Page 8 Line 174- Page 8 Line 181:

Moreover, we detected the CDK13 mRNA (Supplementary Fig. 6a-d) and protein expressions (Supplementary Fig. 6e) and found that they were unaffected by siRNAs and overexpression vector of circCDK13. The results demonstrated that siRNAs used in our study didn't degrade CDK13 linear mRNA transcript and circCDK13 couldn't increase expression of CDK13 mRNA and protein by a positive feedback pathway reported before¹⁸. Collectively, the above findings indicated that circCDK13 rather than CDK13 protein played an essential role in regulating the proliferation and migration of HDFs and HEKs.

References

[1] Qi JC, Yang Z, Lin T, et al. CDK13 upregulation-induced formation of the positive feedback loop among circCDK13, miR-212-5p/miR-449a and E2F5 contributes to prostate carcinogenesis. *J Exp Clin Cancer Res.* 2021;40(1):2.

Supplementary Fig. 6

Supplementary Figure 6. CDK13 mRNA and protein expression in HDFs and HEKs is unaffected by circCDK13 expression. (a, b) Knockdown of circCDK13 by siRNA interference technology in HDFs and HEKs does not affect the expression level of CDK13 mRNA (n=3). (c, d) Exogenous enforced overexpression of circCDK13 in HDFs and HEKs fails to alter the expression level of CDK13 mRNA (n=3). (e) The expression of circCDK13 was overexpressed or knocked down in HDFs and HEKs, and then the expression level of CDK13 protein in cells was detected by

western blot. Data are represented as mean \pm SD. Statistical significance is indicated as follows: NS, not significant.

5. The concentration and duration of cycloheximide treatment were not clear stated in the Methods or Figure legends section.

Response: Thank you for your suggestion. We have added the concentration and duration of cycloheximide treatment in the Figure legend section of Fig. 3t on Page 37 Line 805-Line 807.

Page 37 Line 805-Line 807:

Fig. 3t: The protein level of IGF2BP3 in HDFs and HEKs with knockdown or overexpression of circCDK13 treated with cycloheximide (25 μ g/ml) for 12 h.

6. It is stated in line 271 that “circCDK13 and IGF2BP3 synergistically enhanced the stability of CD44 and c-MYC mRNA”. I wonder whether the authors examined whether circCDK13 or IGF2BP3 directly bind with CD44 and c-MYC mRNA. If not, how circCDK13 and IGF2BP3 affected the stability of CD44 and c-MYC mRNA? Is there any mediator involved in?

Response: Thank you for your question. To further prove that circCDK13 and IGF2BP3 synergistically promote the expression of CD44 and c-MYC by enhancing the stability of CD44 and c-MYC mRNA, we performed the PCR analysis of pull-down products precipitated by circCDK13 probe or IGF2BP3 antibody with the aim of providing the possibility for circCDK13 and IGF2BP3 to bind with CD44 and c-MYC

mRNA. As expected, we found that there was an enrichment of CD44 and c-MYC mRNA in the pull-down products (Supplementary Fig. 14). Previous researches have reported that IGF2BP3 directly bind with CD44 and c-MYC mRNA for stabilization [1,2]. Combining literature reports and our results, we speculated that circCDK13 and IGF2BP3 synergistically promoted the expression of CD44 and c-MYC by enhancing the stability of CD44 and c-MYC mRNA in HDFs and HEKs. The content of this part was described on Page 13 Line 278- Page 14 Line 297.

Page 13 Line 278- Page 14 Line 297:

To further prove that circCDK13 and IGF2BP3 synergistically promote the expression of CD44 and c-MYC by enhancing the stability of CD44 and c-MYC mRNA, we performed the PCR analysis of pull-down products precipitated by circCDK13 probe or IGF2BP3 antibody. As expected, we found that there was an enrichment of CD44 and c-MYC mRNA in the pull-down products (Supplementary Fig. 14), which provided the possibility for circCDK13 and IGF2BP3 to bind with CD44 and c-MYC mRNA for stabilization. However, it's unclear whether circCDK13 or IGF2BP3 directly binds to CD44 and c-MYC mRNA and whether other mediators are involved in the binding of circCDK13-IGF2BP3 complex to CD44 and c-MYC mRNA. One study reported that IGF2BP3 could increase c-MYC mRNA stability by binding to the coding region instability determinant (CRD) residing in the 3'-terminus of the c-MYC coding region²⁶. In another study, IGF2BP3 was found to bind to the 3'-UTR of CD44 mRNA for its stabilization²⁹. Combining literature reports and our results, we speculated that circCDK13 and IGF2BP3 synergistically promoted the expression of CD44 and c-

MYC by enhancing the stability of CD44 and c-MYC mRNA in HDFs and HEKs. Furthermore, we observed significantly lower expression levels of IGF2BP3, CD44, and c-MYC proteins in epidermal keratiocytes and dermal fibroblasts from the DFU group compared with the NWs group (Fig. 4i). Based on the above experimental results, we inferred that circCDK13 and IGF2BP3 synergistically enhance the stability of CD44 and c-MYC mRNA, increase CD44 and c-MYC protein levels, and thereby enhance the proliferation and migration of HDFs and HEKs.

[1] Huang H, Weng H, Sun W, et al. Publisher Correction: Recognition of RNA N6-methyladenosine by IGF2BP proteins enhances mRNA stability and translation. *Nat Cell Biol.* 2020;22(10):1288.

[2] Vikesaa J, Hansen TV, Jønson L, et al. RNA-binding IMPs promote cell adhesion and invadopodia formation. *EMBO J.* 2006;25(7):1456-1468.

Supplementary Fig. 14:

Reviewer #2:

This study aims to identify a novel circular RNA (circRNA) loaded into small extracellular vesicles (sEVs) to reverse the impaired wound healing caused by diabetes.

Utilizing an external database, the investigators identified circCDK13 that was

downregulated in human diabetic wounds. In support of circCDK13, they show depletion of circCDK13 reduces the migration and proliferation of human dermal fibroblasts (HDF) and human epidermal keratinocytes, and overexpression showed the opposite. At a mechanistic level, circCDK13 interacted directly with IGF2BP3 in an m⁶-methyladenosine manner to increase the expression of CD44 and c-Myc. Encapsulation of mesenchymal stem cell-derived small extracellular vesicles (sEVs) that overexpress circCDK13 (circCDK13OE-sEVs) reversed the effects of experimental hyperglycemia (Age-BSA) on the migration and proliferation of HDFs and HEKs. circCDK13OE-sEVs were more effective than N-sEVs in accelerating wound healing and skin appendage regeneration in db/db type 2 diabetic rats and Streptozotocin (STZ)-induced type I diabetic rats.

The experiments showing how circCDK13 interacts with IGF2BP3 and their effects on HDFs and HEKs under hyperglycemic conditions were elegant and convincing. While the circCDK13^{OE}-sEVs appeared to increase the wound healing rate, the design and interpretation raised concerns. Given that the primary goal of these experiments was the development of a therapeutic to reverse wound healing in diabetics, these experiments need to be more detailed.

Response: Thanks very much for your high evaluation.

1. It is stated that circCDK13 was downregulated in “diabetic wounds compared with acute wounds”. Do you mean non-diabetic wounds, or are you referring to acute or chronic diabetic wounds?

Response: Thank you for your question. In this study, “acute wounds” refer to non-diabetic wounds. To avoid confusion, we have replaced “acute wounds” with “non-diabetic wounds” in the abstract on Page 2 Line 36.

2. The optimal experimental wound is stented because rats have a subcutaneous muscle, the panniculus carnosum, that causes the wound to heal by contraction. Stenting the wound promotes a healing pattern more typical of that seen in humans.

Response: Thank you for your suggestion. Yes, it’s very reasonable to stent the experimental wounds for promoting a healing pattern more typical of that seen in humans. In our study, rubber rings (Fig. 6b, Fig. 7b) were attached to the edge of the wounds, which effectively prevented the contraction of the wounds.

Fig. 6b

Fig. 7b

3. None of the wounding experiments were performed to complete healing, which is the proper standard. This is important for several reasons. The primary goal of these experiments is to evaluate a therapeutic intervention. The acceleration of the wound healing rate measured in days is the benchmark to determine if an intervention justifies its cost. None of the experiments contained a wild-type rat

control group.

Response: Thank you for your valuable suggestion. As the proper standard, a control group which complete healing should be specifically designed in the wounding experiments. As you said, the acceleration of the wound healing rate measured in days is the benchmark to determine if an intervention justifies its cost. As you required, we added the supplementary experiment to observe normal wound healing in a wild-type rat with or without circCDK13^{OE}-sEVs. Two full-thickness cutaneous wounds were created on the back of each wild-type rat, followed by subcutaneous injection of circCDK13^{OE}-sEVs, N-sEVs, or an equal volume of PBS. Consistent with the results from the diabetic wound healing, the wound closure in normal wounds was also accelerated by circCDK13^{OE}-sEVs, as determined by the smaller wound areas measured on days 7 and 14 post-wounding compared with N-sEVs and PBS groups (a-d). H&E staining was conducted to evaluate the wound length in each experimental group and the results were consistent with the above wound area measurements, showing a shorter wound length in circCDK13^{OE}-sEVs group (e, f). Furthermore, in tissue sections, more skin appendages, such as hair follicles and sebaceous glands, were observed in circCDK13^{OE}-sEVs-treated wounds on day 14 (e). By comparing the data from diabetic rats and wild-type rats, we found that the wound areas of diabetic rats were larger than that of wild-type rats at the same time and with the same treatment.

circCDK13^{OE}-sEVs promote wound healing in wild-type rat.

(a) Representative images of the wound area by different treatments on days 0, 3, 7, and 14, after operation. Scale bar, 1 mm. (b) Simulation plots of the wound closure areas. Scale bar, 1 mm. (c, d) Quantitative evaluation of the wound closure rate (n=6). (e) Representative images of H&E staining of wound sections. Scale bar, 1 mm. (f) Quantitative analysis of wound width (n=6). Data are represented as mean \pm SD. Statistical significance is indicated as follows: *p < 0.05, **p < 0.01, ***p < 0.001.

4. Wound healing progresses between multiple defined stages: proliferative, angiogenic and remodeling. Understanding the role of circCDK13 during these stages would greatly enhance the significance of the experiments.

Response: Thank you for your valuable suggestion. As you said, the wound healing process can be divided into multiple defined stages including inflammatory phase, proliferative phase, and remodeling phase. In our study, we observed the positive role

of circCDK13 in proliferative phase. Especially, we observed that circCDK13 promoted not only the diabetic wound healing but also the regeneration of skin appendages such as hair follicles and sebaceous glands, which enhanced the significance of circCDK13 application in wound treatment. As for the role of circCDK13 in inflammatory phase and remodeling phase needs to be further investigated.

5. The wound healing rate in the streptozocin-induced type 1 diabetes was only carried out for 14 days. Why? According to the protocol, the intervention was administered until day 17. Why exclude that dose? Moreover, the differences in healing rates were very similar in the N-sEVs and the circCDK13^{OE}-sEVs suggesting a minimal therapeutic effect.

Response: Thank you for your suggestion. We are sorry for causing confusion. As you required, the rate of wound healing on the 17th day after the operation has been added in Fig. 7g. The data exhibition form of Fig. 7g may make you feel that the healing rates are similar in N-sEVs and circCDK13^{OE}-sEVs groups, but in fact there are significant differences in healing rates between two groups as shown as histogram on the 14th day (for example) by statistics.

Fig. 7g:

Fig .7g

Histogram on the 14th day:

6. Given the in vivo model, why not measure the expression of circCDK13 during the different stages of wound healing in diabetic and wild-type rats? Or use a siRNA against circCDK13 in wild-type rats?

Response: Thank you for your insightful review and valuable suggestion. As you required, we used the siRNAs against circCDK13 in wild-type rats and measure the expression of circCDK13 during the different stages of wound healing in diabetic and wild-type rats. Two full-thickness skin wounds were created on the back of each wild-type rat and then the siRNAs were used to knock down the expression of circCDK13 in the wound tissue. The results showed that circCDK13 knockdown decreased the wound healing rate, as determined by larger wound areas measured on day 3, 7, and 14 post-wounding compared with control groups (a-e). H&E staining was conducted to evaluate the wound length in each experimental group and the results were consistent with the above wound area measurements (f, g). Furthermore, fewer skin appendages, such as hair follicles and sebaceous glands, were observed in circCDK13 knockdown wounds on day 14 (h). Additionally, the expression of circCDK13 was measured during the different stages of wound healing in diabetic and wild-type rats. At the designed time points, we found that the expression of circCDK13 in the wound tissue of diabetic rats was significantly reduced compared with the wounds of wild-type rats (i). In summary, the decrease of circCDK13 in skin wounds resulted in slow wound healing.

Decrease of circCDK13 in skin wounds resulted in slow wound healing.

(a) Representative images of the wound area by different treatments on days 0, 3, 7, and 14, after operation. Scale bar, 1 mm. (b) Simulation plots of the wound closure areas. Scale bar, 1 mm. (c-e) Quantitative evaluation of the wound closure rate (n=6). (f) Representative images of H&E staining of wound sections on day 7. Scale bar, 1 mm. (g) Quantitative analysis of wound width (n=6). (h) Representative images of H&E staining of wound sections on day 14. Scale bar, 200 μ m. (i) RT-qPCR analysis of circCDK13 expression abundance in normal skin of wild-type rats (Normal skin), wounds of wild-type rats (Normal wound), and wound of diabetic rats (Diabetic wound) at different time points (N=6). Data are represented as mean \pm SD. Statistical

significance is indicated as follows: NS, not significant, * $p < 0.05$, ** $p < 0.01$, *** $p < 0.001$. Compare to Norma skin group, ### $P < 0.001$.

Reviewer #3:

In the present study, author revealed that impaired healing of diabetic wounds was tightly associated with the downregulation of circCDK13, and then confirmed a novel wound healing-promoting mechanism that circCDK13 directly interacted with IGF2BP3 to form a circRNA-protein-mRNA ternary complex, which synergistically enhanced the stability of CD44 and c-MYC mRNA to promote the proliferation and migration of HDFs and HEKs. Author successfully constructed engineered sEVs bearing circCDK13 and corroborated that circCDK13^{OE}-sEVs. Author confirmed that in the wounds of db/db diabetic mice and STZ-induced type I diabetic rats, circCDK13^{OE}-sEVs could accelerate re-epithelialization and granulation tissue formation, and promote wound remodeling and regeneration of skin appendages.

Response: Thanks very much for your high evaluation.

1. The co-localization experiments of circCDK13 and IGF2BP3 using Fish showed that circCDK13 and IGF2BP3 interact. Please provide more data to support this finding.

Response: Thank you for your suggestion. As you required, the binding between circCDK13 and IGF2BP3 was confirmed by RNA pull-down and RIP-qPCR assays (Fig. 3h-j). Moreover, we carried out an RNA FISH-immunofluorescence (FISH-IF)

analysis and found that circCDK13 colocalized with IGF2BP3 in the cytoplasm of HDFs and HEKs (Fig. 3k). All above results showed that circCDK13 and IGF2BP3 interacted with each other.

Fig. 3h

Fig. 3i

Fig. 3j

Fig. 3k

2. The authors perform immunoprecipitation of circCDK13 followed by mass-spectrometry to identify protein interactors. These data are not shown: Fig. S7 just shows the LC-MS/MS plot, while a supplementary table with the identified interactors should be provided.

Response: Thank you for your valuable suggestion. As you required, a supplementary table containing the identified interactors has been supplemented in the Supplementary Material.

3. Authors should examine studies of circCDK13^{OE}-sEVs on function in other

organs (e.g., kidneys, etc.)

Response: Thank you for your valuable suggestion. As you required, we examined whether circCDK13^{OE}-sEVs could function on other organs including the lungs, liver, spleen, kidney, and heart of the rats after multiple administrations of N-sEVs or circCDK13^{OE}-sEVs. H&E staining of the heart, lung, liver, kidney, and spleen in circCDK13^{OE}-sEVs group demonstrated no histological abnormalities or immune cell infiltration in comparison with those treated with N-sEVs or PBS. Therefore, we initially believe that multiple administrations of N-sEVs or circCDK13^{OE}-sEVs to skin wounds are safe *in vivo* and have no negative impact on the vital organs.

Effects of circCDK13^{OE}-sEVs on other organs in rats

Representative images of H&E staining of other organs in rats, such as lungs, liver, spleen, kidney and heart. scale bar, 40 μm .

4. The authors should examine the effect of overexpression or knockdown of circCDK13 on cell death (such as autophagy, apoptosis, necrosis, etc.) in HDFs and HEKs.

Response: Thanks for your valuable suggestion. As you required, we evaluated the effects of circCDK13 on autophagy, apoptosis, and necrosis of HDFs and HEKs with knockdown or overexpression of circCDK13. The results of Monodansylcadaverine (MDC) method showed that circCDK13 did not significantly affect autophagy levels (a-e). Flow cytometric analysis showed that knockdown or overexpression of circCDK13 in HDFs and HEKs did not significantly affect cell apoptosis and necrosis (f).

Effect of CircCDK13 on autophagy, apoptosis, and necrosis in HDFs and HEKs

(a) Monodansylcadaverine (MDC) fluorescent probe was used to detect autophagy levels in HDFs and HEKs transfected with si-NC or si-circCDK13, and vector or circCDK13-OE. scale bar, 40 μ m. (b-e) After the MDC fluorescent probe has been used to label the autophagosome in HDFs and HEKs, the fluorescence intensity at 335/512 nm is measured using a fluorescence microplate reader, and the relative fluorescence intensity is finally calculated (n=6). (f) Flow cytometry analysis of apoptosis and necrosis in HDFs and HEKs transfected with si-NC or si-circCDK13, and vector or

circCDK13-OE. Data are represented as mean \pm SD. Statistical significance is indicated as follows: NS, not significant.

5. Other studies on circCDK13 have shown that activation of endogenous CDK13 can promote the expression of circCDK13, and the high expression of circCDK13 promotes the occurrence and development of prostate cancer, while inhibiting the expression of circCDK13 can reduce the occurrence of prostate cancer. Authors should consider whether the expression of circCDK13 might produce tumorigenicity.

Response: Thank you for your insightful review and valuable comments. Qi et al. found that CDK13 is significantly upregulated in human prostate cancer (PCa) tissue[1]. CDK13 depletion and overexpression in PCa cells decrease and increase, respectively, cell proliferation, and the pro-proliferation effect of CDK13 is strengthened by its interaction with E2F5[1]. Mechanistically, transcriptional activation of endogenous CDK13, but not the forced expression of CDK13 by its expression vector, remarkably promotes E2F5 protein expression by facilitating circCDK13 formation. Further, the upregulation of E2F5 enhances CDK13 transcription and promotes circCDK13 biogenesis, which in turn sponges miR-212-5p/449a and thus relieves their repression of the E2F5 expression, subsequently leading to the upregulation of E2F5 expression and PCa cell proliferation [1].

In this study, we found that circCDK13 expression was significantly downregulated in diabetic wound tissue compared to normal wound tissue, and

confirmed that circCDK13 depletion and overexpression in HDFs and HEKs decrease and increase, respectively, cell proliferation and migration. Therefore, we hypothesized that the delivery of circCDK13 by sEVs would be able to compensate for the lack of circCDK13 in HDF and HEK in diabetic wounds, thereby promoting diabetic wound healing. Changes in the expression level of circCDK13 in cells were analyzed by RT-qPCR when circCDK13^{OE}-sEVs was co-incubated with HDFs or HEKs. We found that intracellular circCDK13 levels increased significantly after 6 h and 12 h of exposure to circCDK13^{OE}-sEVs, and then the abundance of circCDK13 gradually decreased over time (a, b). The abundance of circCDK13 was close to that in control cells after 120 h (a, b). Exogenous circRNAs can be degraded in cells, although circRNAs is more stable than linear RNAs.

In addition, we found that knockdown or overexpression of circCDK13 in HDFs and HEKs did not alter the expression levels of CDK13 protein (Supplementary Fig. 6e), thus not forming the CDK13-circCDK13-miR-212-5p/449a-E2F5 positive feedback loop to overstimulate cell proliferation and migration. Therefore, we are confident that the administration of exogenous circCDK13 for the treatment of diabetic wounds is safe and does not lead to tumourigenesis.

circCDK13 expression over time in HDFs and HEKs co-incubated with circCDK13^{OE}-sEVs

(a, b) After co-incubation of circCDK13^{OE}-sEVs with HDFs or HEKs, cells were harvested at defined time points, total RNA was extracted, and then the expression level of circCDK13 was analyzed by RT-qPCR (n=3). Data are represented as mean ± SD. Statistical significance is indicated as follows: NS, not significant; **p < 0.01, ***P < 0.001.

Supplementary Fig. 6c:

[1] Qi JC, Yang Z, Lin T, et al. CDK13 upregulation-induced formation of the positive feedback loop among circCDK13, miR-212-5p/miR-449a and E2F5 contributes to prostate carcinogenesis. J Exp Clin Cancer Res. 2021;40(1):2.

6. The authors should increase the detection of whether circCDK13 affects classical proliferation pathways (such as Yap).

Response: Thanks for your valuable suggestion. As you required, we performed the detection of whether circCDK13 affects YAP pathway. In HDFs and HEKs, we observed that the knockdown of endogenous circCDK13 increased YAP phosphorylation levels, while upregulation of circCDK13 reduced YAP phosphorylation levels, but did not alter YAP protein expression levels.

Effects of circCDK13 overexpression or knockdown on YAP pathway

(a, b) The protein levels of p-YAP and YAP were detected by western blot assay in HDFs and HEKs transfected with si-NC or si-circCDK13, and vector or circCDK13-OE.

7. Macrophages play an indispensable role in the process of wound repair in the body. The authors should examine the expression of circCDK13 and IGF2BP3 in macrophages and its influence on them.

Response: Thanks for your valuable suggestion. As you required, we examined the expressions of circCDK13 and IGF2BP3 in macrophages and its influence on them. The initial macrophages (M0) were induced to M1 macrophages or M2 macrophages,

respectively. The expressions of circCDK13 and IGF2BP3 were detected in M0, M1, and M2 macrophages. Finally, we did not observe any significant difference in the levels of expression of IGF2BP3 (a) and circCDK13 (b) in macrophages with different phenotypes.

Expression levels of circCDK13 and IGF2BP3 in macrophages with different phenotypes

(a) The protein levels of IGF2BP3 were detected by western blot assay in macrophages with different phenotypes. (b) RT-qPCR analysis of circCDK13 abundance in macrophages with different phenotypes (n=3). Data are represented as mean \pm SD. Statistical significance is indicated as follows: NS, not significant.

8. Most chronic wounds are characterized by a large number of inflammatory cell infiltration, leading to overexpression of reactive oxygen species and MMP. It is suggested that the authors examine the effects of overexpression or knockdown of circCDK13 on the expression of reactive oxygen species and MMP.

Response: Thank you for your valuable suggestion. In this study, we found that the expressions of reactive oxygen species (ROS), MMP2, and MMP9 were not significantly affected by knockdown or overexpression of circCDK13 in HDFs (a-c). When circCDK13 was knocked down in HEKs, the expression levels of MMP2 and

MMP9 were slightly decreased (d), while when circCDK13 was overexpressed, the expression levels of MMP2 and MMP9 were slightly increased (d), but the levels of ROS did not change significantly in HEKs (e, f).

Effects of circCDK13 overexpression or knockdown on reactive oxygen species and MMP expressions

(a) The protein levels of MMP9 and MMP2 were detected by western blot assay in HDFs transfected with si-NC or si-circCDK13, and vector or circCDK13-OE. (b, c) The expression levels of reactive oxygen species were measured using a fluorescence microplate reader in HDFs transfected with si-NC or si-circCDK13, and vector or circCDK13-OE (n=6). (d) The protein levels of MMP9 and MMP2 were detected by western blot assay in HEKs transfected with si-NC or si-circCDK13, and vector or circCDK13-OE. (e, f) The expression levels of reactive oxygen species were measured

using a fluorescence microplate reader in HEKs transfected with si-NC or si-circCDK13, and vector or circCDK13-OE (n=6). Data are represented as mean \pm SD. Statistical significance is indicated as follows: NS, not significant.

9. Please revise the English grammar and writing style of the manuscript, spelling and grammatical errors should be excluded.

Response: Thanks for your suggestion. We are very sorry for the spelling and grammatical errors. To revise the English grammar and writing style of the manuscript, we asked our colleague whose native language is English to review our revised manuscript. We hope our new manuscript will meet the requirement for publication.

REVIEWERS' COMMENTS

Reviewer #1 (Remarks to the Author):

I am satisfied with the response.

Reviewer #2 (Remarks to the Author):

Despite my request, the additional wound healing experiments that yielded significant new information, only the wild type rat achieved 100% healing of wound. Nonetheless, the other concerns were adequately addressed.

Reviewer #3 (Remarks to the Author):

In the present study, author revealed that impaired healing of diabetic wounds was tightly associated with the downregulation of circCDK13, and then confirmed a novel wound healing-promoting mechanism that circCDK13 directly interacted with IGF2BP3 to form a circRNA-protein-mRNA ternary complex, which synergistically enhanced the stability of CD44 and c-MYC mRNA to promote the proliferation and migration of HDFs and HEKs.

Author successfully constructed engineered sEVs bearing circCDK13 and corroborated that circCDK13OE-sEVs, and confirmed that in the wounds of db/db diabetic mice and STZ-induced type I diabetic rats, circCDK13OE-sEVs could accelerate re-epithelialization and granulation tissue formation, and promote wound remodeling and regeneration of skin appendages.

The authors addressed all the issues, no more comments.

1) Reviewer #1 (Remarks to the Author):

Comments: I am satisfied with the response.

Response: Thank you for your insightful suggestion. We appreciate your time and efforts in reviewing our manuscript and are glad that the revisions have met your expectations.

2) Reviewer #2 (Remarks to the Author):

Comments: Despite my request, the additional wound healing experiments that yielded significant new information, only the wild type rat achieved 100% healing of wound. Nonetheless, the other concerns were adequately addressed.

Response: Thank you for your comment on the result of the additional wound healing experiments on the wild type rat. As you said, the primary goal of the wounding experiments is to evaluate the therapeutic efficacy of circCDK13^{OE}-sEVs and the acceleration of the wound healing rate measured in days is important. So, we gave the complete healing time of the wounds on the db/db diabetic mice (a) and STZ-induced type I diabetic rats (b).

3) Reviewer #3 (Remarks to the Author):

Comments: In the present study, author revealed that impaired healing of diabetic wounds was tightly associated with the downregulation of circCDK13, and then confirmed a novel wound healing-promoting mechanism that circCDK13 directly interacted with IGF2BP3 to form a circRNA-protein-mRNA ternary complex, which synergistically enhanced the stability of CD44 and c-MYC mRNA to promote the

proliferation and migration of HDFs and HEKs.

Author successfully constructed engineered sEVs bearing circCDK13 and corroborated that circCDK13OE-sEVs, and confirmed that in the wounds of db/db diabetic mice and STZ-induced type I diabetic rats, circCDK13^{OE}-sEVs could accelerate re-epithelialization and granulation tissue formation, and promote wound remodeling and regeneration of skin appendages.

The authors addressed all the issues, no more comments.

Response: Thank you for your affirmation of our research. Your feedback has provided us with valuable theoretical guidance and enriched the content of our study.